

# Countrywide Digital Surface Models and Vegetation Height Models from Historical Aerial Images

Mauro Marty[1], Livia Piermattei[1,2], Lars T. Waser[1], Christian Ginzler[1]

[1] Land Change Science Research Unit, Swiss Federal Institute for Forest, Snow and Landscape Research WSL, Birmensdorf, 8903, Switzerland

[2] Department of Geography, University of Zurich, Zurich, 8057, Switzerland

*Correspondence to*: Mauro Marty (mauro.marty@wsl.ch)

**Abstract.** Historical aerial images, captured by film cameras in the previous century, are valuable resources for quantifying Earth's surface and landscape changes over time. In the post-war period, these images were often acquired to create topographic maps, resulting in the acquisition of large-scale aerial photographs with stereo coverage. Photogrammetric techniques applied to these stereo images enable the extraction of 3D information to reconstruct digital surface models (DSMs) and orthoimages.

Here, we present a highly automated photogrammetric approach for generating countrywide DSMs of Switzerland, at a 1 m resolution, from approximately 40,000 scanned aerial stereo images acquired between 1979 and 2006, with known exterior and interior orientation. We derived four countrywide DSMs for the epochs 1979–1985, 1985–1991, 1991–1998, and 1998–2006. From the DSMs, we generated corresponding countrywide vegetation height models (VHMs). We assessed the quality of the historical DSMs at the country scale and within six representative study sites, evaluating the vertical accuracy and the completeness of image-matching across different land cover types.

Mean completeness ranged from 64% for 'glacial and perpetual snow' to 98% for 'sealed surfaces', with a value of 93% for the 'closed forest' class. Across Switzerland, the median elevation accuracy of the historical DSMs compared with a reference digital terrain model (DTM) on sealed surface points ranged from 0.28 to 0.53 m, with a normalised median absolute deviation (NMAD) of around 1 m and a maximum root mean square error (RMSE) of 3.90 m. The same analysis between geodetic points and historical DSMs showed higher accuracies, with median values of ≤ 0.05 m and an NMAD < 1 m.

The VHMs generated in this study enabled the detection of major changes in forest areas due to windstorm damage, forest dynamics, and growth. This work demonstrates the feasibility of generating accurate, very high-resolution DSM time series (spanning three decades) and VHMs from historical aerial images of the entire surface of Switzerland in a highly automated manner. The VHMs are already being used to estimate countrywide biomass changes. The countrywide DSMs and VHMs for the four epochs, along with auxiliary data, are available online at https://doi.org/10.16904/envidat.528 (Marty et al., 2024) and can be used to quantify long-term elevation changes and related processes across different surfaces.



## 1 Introduction

Photographs taken by film cameras on aeroplanes throughout the last century represent a unique and invaluable information source for quantifying past changes in the Earth's surface and landscape. Various institutions and mapping agencies have recognised this value by preserving and scanning these photographs and making them publicly available.

After the Second World War, large-scale aerial photographs were frequently taken to create topographic maps. To ensure stereoscopic coverage and enable the extraction of three-dimensional (3D) information, image blocks were acquired in parallel strips, with each photograph overlapping the next along the same flight line (forming a stereo pair). Due to the stereoscopic acquisition and very high resolution of these images, digital surface models (DSMs) and orthoimages at a metric or sub-metric resolution can be generated using digital photogrammetry and computer vision techniques. This capability has led to the use of historical aerial images to assess long-term surface elevation changes across various geoscience fields.

Multi-temporal DSMs derived from historical aerial images have frequently been used in cryospheric research for assessments of changes in glacier elevation (e.g., Magnússon et al., 2016; Belart et al., 2020; Denzinger et al., 2021), in periglacial studies (e.g., Cusicanqui et al., 2021; Fleischer et al., 2021), in geomorphology (e.g., Micheletti et al., 2015; Piermattei et al., 2023; Schwat et al., 2023), in volcanology (e.g., Gomez et al., 2015), in analyses of land-use change (e.g., Nebiker et al., 2014; Bolles et al., 2018) and in archaeology (e.g., Risbøl et al., 2015; Peppa et al., 2018). Most studies have been focused on local scales and have involved processing a small set of scanned aerial images, with a few considering larger scales (Geyman et al., 2022; Muhammed et al., 2023) or even regional scales such as Greenland (Korsgaard et al., 2016).

In this study, we generated countrywide DSMs of Switzerland using historical aerial images acquired from the 1980s to the early 2000s. Additionally, as we carried out the study within the framework of the Swiss National Forest Inventory (NFI), we produced corresponding countrywide vegetation height models (VHMs). The Swiss NFI aims to collect objective information on forests at the national level for scientific, political, social and economic purposes (Abegg et al., 2023). Data collection within the context of the NFI is mainly based on a systematic terrestrial sample inventory, which is intended to provide results for Switzerland as a whole, as well as for larger regions and individual cantons. Since the early days of the NFI, aerial image interpretation and remote sensing data have been used. Over the decades, the image data has evolved from analogue black-and-white images to digital RGB true colour aerial images, and finally to the digital colour infrared (CIR) sensor data used today. In addition to the application of stereo-image-based analysis on various sample grids, comprehensive datasets have been developed and produced.

The use of historical aerial images for 3D forest reconstruction is limited, with most studies focused on 2D changes in tree and canopy cover at local scales using orthoimages (e.g., Kadmon and Harari-Kremer, 1999; Morgan and Gergel, 2010; Kulha et al., 2018) and texture analysis (Hudak and Wessman, 1998; Kupidura et al., 2019). In Central Switzerland, Waser et al. (2008a, b) used scanned colour infrared aerial images to generate DSMs and quantify tree/shrub cover changes between 1997 and 2002. Recent research has focused on methods based on artificial intelligence (AI) to extract information on woody vegetation along alpine treeline ecotones, tree cover (Wang et al., 2022), and forest cover (Hufkens et al., 2020) from historical aerial



orthoimages. The study by Véga and St-Onge (2008) was among the first where forest canopy growth was quantified using
DSM time series from historical aerial images. As changes in the height of forest canopies over time represent a key aspect of
forest dynamics, further studies have been conducted to explore the use of historical images for tracking long-term changes in
canopy height (Nurminen et al., 2015; Bożek et al., 2019; Hufkens et al., 2020) and forest stand age (Vastaranta et al., 2015)
and for describing the evolution of forest successional stages in tropical forests (Berveglieri et al., 2016, 2018).
Here, we present a highly automated approach to process scanned aerial images (hereafter referred to as 'historical aerial
images') for generating countrywide DSMs and VHMs in Switzerland. Expanding on the work by Heisig and Simmen (2021),
who generated a single countrywide orthoimage of Switzerland from images taken between 1985 and 1991, we processed all
scanned images available from swisstopo, the Swiss Federal Office of Topography (swisstopo, 2024a), acquired over
Switzerland between 1979 and 2006. Countrywide coverage was obtained using image acquisition intervals of approximately
seven years, leading to four countrywide DSM products for the epochs 1979–1985, 1985–1991, 1991–1998, and 1998–2006.
To derive VHMs, we normalised the elevation values from the historical DSMs by subtracting the available countrywide
digital terrain model (DTM; swissALTI3D version 2017, swisstopo). We assessed the quality of the generated DSMs by
evaluating the completeness of the image-matching process across different land cover types and the vertical accuracy of the
historical DSMs by comparing elevation values with: (i) the reference DTM, (ii) geodetic elevation points on a national scale,
and (iii) manual stereoscopic elevation measurements from six representative study sites in Switzerland. We close the article
by highlighting the potential applications of these datasets in research on forest dynamics and management.
**2 Dataset and methods**
**2.1 Historical aerial stereo imagery of Switzerland**
DSMs and VHMs were generated from historical images covering the entire 41,285 km$^2$ of Switzerland, with an elevation
range between 197 and 4634 m a.s.l. In the Swiss NFI, Switzerland is characterised by different types of landscapes and is
divided into five main forest production regions: the Jura in the northwest (1), the Swiss Plateau (2), the Prealps (3) between
the Swiss Plateau and the Alps, the Alps (4/4a), and the Southern Alps (5) (Fig. 1). These regions exhibit different climatic
and topographic characteristics, such as land cover classes, elevation ranges, tree species and forest structures (Tab. 1).
Representative study sites of approximately 210 km$^2$ (i.e., one map sheet; see section 2.2.2) were selected within these regions
(Fig. 1). In the Alps region, an additional study site encompassing the tongue of the Grosser Aletsch Glacier was included.
The data used to generate the DSMs consists of approximately 40,000 panchromatic images acquired with an RC10 aerial
camera manufactured by Wild Heerbrugg (currently owned by Leica Geosystems, Heerbrugg, Switzerland). Multiple airborne
campaigns were conducted from 1979 to 2006, with each campaign spanning approximately seven years to cover the entire
country. The first countrywide coverage was obtained from the acquisition campaign that took place between 1979 and 1985
(epoch 1), followed by subsequent campaigns from 1985 to 1991 (epoch 2), 1991 to 1998 (epoch 3), and 1998 to 2006 (epoch
4) (Fig. 2).

**Figure 1: The Swiss NFI forest production regions (1 = Jura, 2 = Swiss Plateau, 3 = Pre-Alps, 4 = Alps, 5 = Southern Alps) in Switzerland and the respective location of the study site for each region, along with the generated orthoimage for epoch 1 (1979–1985). An additional study site (4a) was selected for the Alps forest production region.**

The primary purpose of these photograph acquisitions was to update the topographic maps produced by swisstopo. The photographs were acquired with approximately 70–80% front overlap and 30–40% side overlap. The average flight height for all datasets was approximately 4,000 m above the ground, with an image scale of about 1:25,000. The photographs were scanned with a resolution of 14 µm using a Leica DSW700 scanner (swisstopo). This resulted in an average ground sample distance (GSD) of around 0.35 m in the scanned images, with larger variations in high-elevation areas. Historical aerial images are freely available on swisstopo's website (swisstopo, 2024a) and are provided with exterior orientation parameters, as well



as camera calibration protocols containing interior orientation parameters such as focal length and fiducial mark values. Figure
2 shows the image footprints of the stereo pairs used to generate the DSMs, along with the corresponding year of acquisition.
**Table 2: Forest production region characteristics. Topographic parameters are based on the swissALTI3D digital terrain model**
**(DTM) version 2017. Forest parameters are based on the Swiss National Forest Inventory (NFI).**

| Forest production region | | Area | Elevation range | Mean slope | Forest cover | Forest type | | | |
|---|---|---|---|---|---|---|---|---|---|
| No. | Name | (km²) | (m a.s.l.) | (°) | (%) | pure coniferous (%) | mixed coniferous (%) | mixed deciduous (%) | pure deciduous (%) |
| 1 | Jura | 4935.4 | 244–1677 | 12.3 | 39.5 | 28.6 | 25.5 | 18.8 | 22.3 |
| 2 | Swiss Plateau | 9412.8 | 311–1685 | 7.2 | 24.2 | 37.8 | 22.9 | 13.6 | 22.1 |
| 3 | Prealps | 6608.1 | 370–2899 | 19.8 | 32.5 | 53.7 | 22.4 | 8.2 | 9.6 |
| 4 | Alps | 16782.6 | 370–4634 | 28.9 | 21.0 | 71.0 | 9.6 | 5.8 | 8.6 |
| 5 | Southern Alps | 3545.7 | 192–3899 | 31.6 | 39.2 | 31.7 | 7.4 | 5.2 | 51.9 |



**Figure 2: Stereo footprint coverage over Switzerland for the four epochs and the number of images used for the respective digital surface model (DSM) generation (barplot, bottom). There is an overlap of acquisition years between adjacent epochs, but images from the same year cover different forest production regions in each epoch.**

## 2.2 DSM and VHM generation

The scanned input images, auxiliary data, and the steps used to generate the DSMs and the VHMs are illustrated in Fig. 3. The countrywide DSMs and VHMs are provided at 1 m spatial resolution in the Swiss projected coordinate system CH1903+ / LV95 (EPSG code 2056), and height information is supplied in LN02 (EPSG code 5728) where applicable. The scanned historical images were processed using the commercial photogrammetric software SocetSet (v5.6.0) by BAE Systems (Falls



Church, USA). With known interior and exterior orientation parameters, DSMs can be generated using the Next-Generation
Automatic Terrain Extraction (NGATE) package implemented in SocetSet (DeVenecia et al., 2007). The exterior orientation
information for the images from epochs 1, 2 and 4 was derived by swisstopo within the framework of the Swiss Land
Use/Cover Statistics program around 2004. Image orientation was performed using ground control points (GCPs), visible in
the historical imagery, and in more recent already-oriented digital stereo images available from swisstopo. The elevation of
the GCPs was stereoscopically extracted from the digital images. This image orientation resulted in accurate horizontal
positioning but biases in the Z direction, ranging from 1 to 5 m, randomly distributed across Switzerland. Therefore, the DSMs
from these epochs were vertically corrected, as described in section 2.2.2. In contrast, the images for epoch 2 were recently
oriented using a highly automated workflow, as described by Heisig and Simmen (2021).

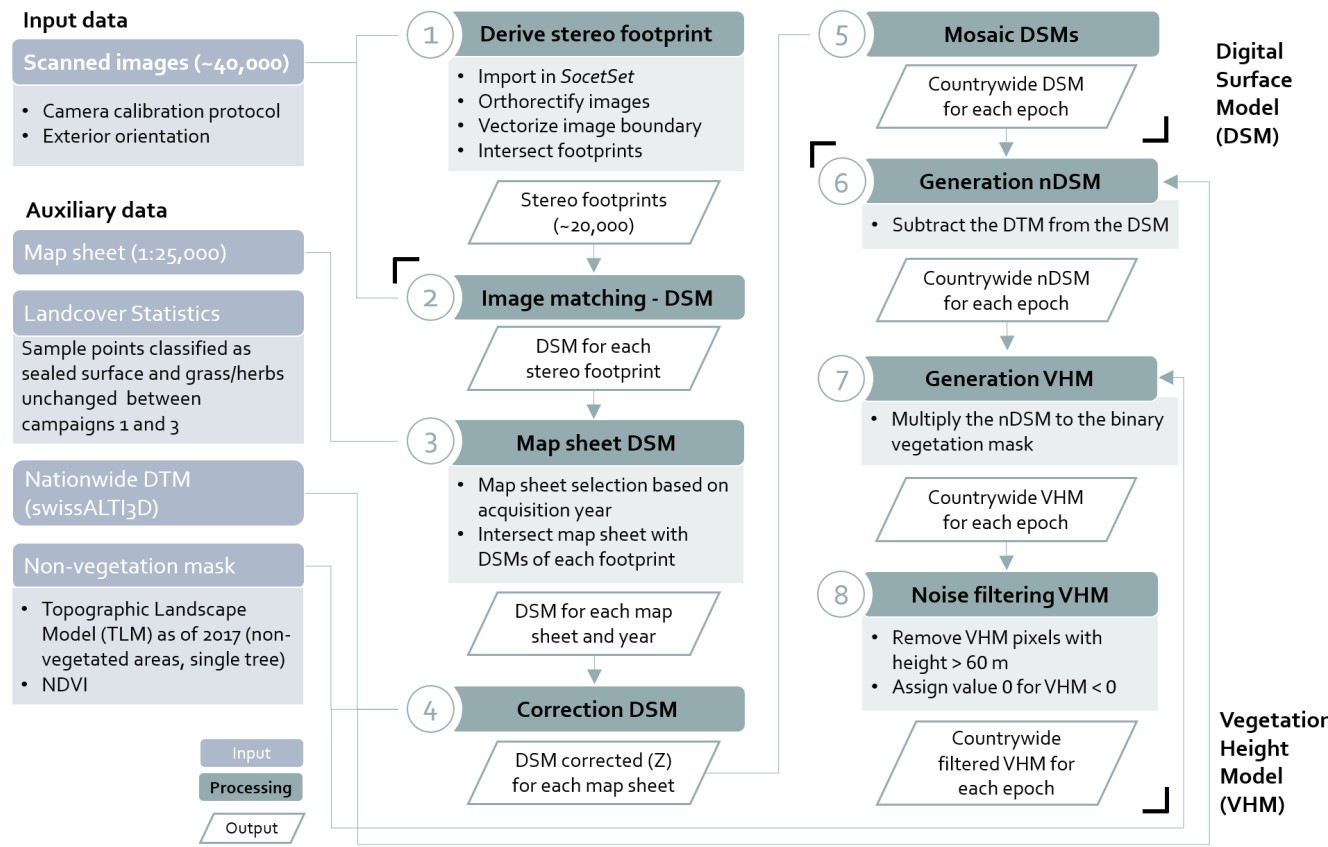

**Figure 3: Workflow of historical image processing and countrywide digital surface model (DSM) and vegetation height model (VHM)**
**generation.**
**2.2.1 Image-matching and DSMs**
The DSM generation process using the NGATE package followed a highly automated workflow. To handle the DSM
reconstruction process for approximately 40,000 images, image-matching was performed on distributed machines for single



stereo footprints (i.e., stereo pairs). In the initial step, the footprint for each image was calculated, and the footprints of adjacent
images along the same flight line were intersected to create stereo footprints. In total, 20,346 stereo footprints were processed.
All stereo footprints were stored in a table on a PostGIS database, which was used to control the image-matching process
running on distributed virtual machines. Since stereo pairs were formed only from the neighbouring images along the same
flight line, multi-image-matching was not involved in this process.
NGATE provides several predefined correlation/matching strategies for different types of image content, such as urban, flat
and low-contrast areas. To ensure the extraction of accurate and complete elevation information across different land cover
types, two strategies were adopted, namely urban and low contrast. The urban strategy utilises internal correlation parameters
that account for elevation discontinuities, while the low-contrast strategy aims to identify congruent points on highly
homogeneous surfaces. While image-matching in NGATE is conducted on every pixel of the input imagery, the resulting
output file represents a regular grid of three-dimensional (XYZ) coordinates. Optionally, this grid can include information
about successful image-matching or interpolation for the given pixel, as indicated by the Figure of Merit value, a product
derived by the software. The spatial resolution of the output DSM is user-defined and, in our case, was set to 1 m, approximately
three times the GSD of the scanned images.
The image-matching using the two strategies resulted in two separate DSMs for each stereo footprint. These two DSMs were
combined into one DSM by including all successfully matched points from the urban strategy, as forests typically exhibit large
elevation discontinuities. Any remaining gaps in this DSM were filled with correlated points from the low-contrast strategy,
where available.

### 2.2.2 DSM merging and vertical correction

The exterior orientation of the images from epochs 1, 3 and 4 resulted in a vertical offset of up to 5 m between the DSMs and
the reference DTM on stable terrain (i.e., sealed surfaces and grass/herb points; see section 2.3.1). Therefore, the DSMs from
these epochs were vertically corrected. For consistency, this correction was also applied to the DSM from epoch 2, resulting
in neglectable changes due to the already precise horizontal and vertical orientation.
The correction of approximately 20,000 DSMs of single stereo footprints was performed at the level of 1:25,000 map sheet
units, provided by swisstopo (swisstopo, 2024b). These map sheets were originally used to plan airborne image acquisition
campaigns. For each map sheet geometry, all DSMs of single stereo footprints belonging to the same epoch were selected via
spatial intersection, merged and cut to the map sheet extent (approximately 50 stereo footprints in flat areas and up to 150 in
high-alpine regions). Since adjacent stereo footprints always overlapped spatially, pixels on these locations often included
multiple elevation values, which were generalised using the median height. Grid cells without any successfully matched
elevation information were interpolated using a triangular irregular network (TIN) surface.
The vertical correction of the generated DSM for each map sheet and epoch is illustrated in Fig. 4. The workflow consisted of
calculating the difference between the reference DTM (swissALTI3D version 2017; see section 2.2.3) and the generated DSM
at points corresponding to the classes 'sealed surface' and 'grass/herb' in the first and third campaigns of the Swiss Land





Use/Cover Statistics program (1979–1985 and 2004–2009, respectively; see section 2.3.1). With this approach, only points
with an unchanged land cover classification were used. Outliers were filtered out based on mean ± 1 standard deviation (Std),
calculated in a 50 × 50 pixel moving window. The median of the remaining elevation differences for each of the 1200 regular
grid cells per map sheet was then calculated. Finally, a coarse correction surface (spline interpolation) was derived from these
median values and applied to the resulting DSMs at the map sheet unit level to correct elevation bias.

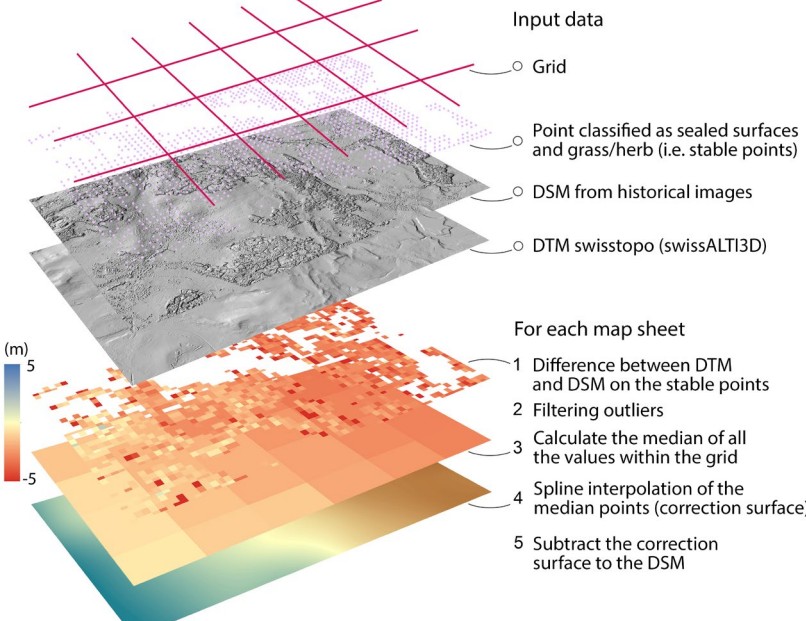


**Figure 4: Illustration of the workflow to correct the elevation bias in the historical digital surface models (DSMs) for each map sheet.**
**The correction is based on the elevation difference between the reference digital terrain model (DTM) and the historical DSM for**
**sample points classified as grass/herb or sealed surfaces.**

**2.2.3 Countrywide DSMs and VHMs**
After randomly distributed elevation biases (section 2.2.2) were corrected in each map sheet DSM of each epoch, the individual
map sheet DSMs were combined into one DSM as a Geotiff file, covering the whole of Switzerland.
The four countrywide VHMs were derived by calculating the difference between the historical DSMs and the reference DTM
(swissALTI3D version 2017, swisstopo; see workflow in Fig. 3). This process resulted in a normalised DSM (nDSM),
representing the heights of all existing objects above ground. Note that the reference DTM was resampled from the default
resolution of 2 m to 1 m to match the resolution of the generated DSMs. The swissALTI3D DTM is based on airborne laser
scanning data for elevations up to 2000 m a.s.l., with an overall point density of 0.5 points/m$^2$ and a vertical accuracy of ±0.5
m (standard deviation). Above 2000 m a.s.l., elevation information is derived by image-matching and manual editing, yielding
a documented vertical accuracy of 1–3 m on average (swisstopo, 2024c).



nDSM raster cells with values > 60 m were considered outliers due to erroneous image-matching and thus were replaced by
'no data'. Next, the VHM (i.e., the nDSM of vegetated areas) was generated by assigning a height above ground of 0 m to all
non-vegetated objects. To achieve this, a binary non-vegetation mask (0 = non-vegetation and 1 = vegetation) was created by
combining information from the topographic landscape model (TLM; swisstopo, 2024d) and a normalised difference
vegetation index (NDVI) map obtained from aerial orthoimages of the years 2009–2015. All areas classified in the TLM as
building, street, and water surface, as well as pixels with an NDVI < 0.1, were set to 0. To retain vegetation close to buildings
and streets in the VHMs, single trees and hedges from the TLM were buffered by 15 m and set to 1. This non-vegetation mask
was multiplied by the nDSMs. The resulting VHMs had a spatial resolution of 1 m, consistent with the historical DSMs.
**2.3 Quality assessment of the DSMs**
The quality of the generated DSMs was assessed by analysing: (1) the completeness achieved through the image-matching
process over different land cover types and (2) the elevation accuracy based on elevation differences to reference datasets (see
section 2.3.1). The elevation differences were calculated consistently as the reference data minus the historical DSMs.
Therefore, positive differences indicated that historical DSMs had negative biases, i.e., that the reference data was higher than
the historical DSM. Quality assessment was performed and reported for the six selected areas and the countrywide DSMs.
**2.3.1 Reference data**
The reference dataset used to assess the completeness of the image-matching process over different land cover classes consisted
of sample point interpretation on a regular 100 m grid across Switzerland. These sample points were collected by the Swiss
Federal Statistical Office in the framework of the Swiss Land Cover/Use Statistics program (BFS, 2024) (Fig. A1). These
countrywide land cover statistics campaigns took place in 1979–1985, 1992–1997, 2004–2009 and 2013–2018. The
completeness for epoch 1 was assessed, and the reference data from the 1979–1985 campaign was used. This dataset comprised
121,896 points for the 6 selected areas. The 27 specific land cover classes were grouped into 6 categories: (1) sealed surface,
(2) bare land, (3) grass/herb, (4) shrub, (5) closed forest, and (6) glacial and perpetual snow.
The vertical accuracy of the DSMs was assessed by comparing the elevation with three reference datasets. First, the sealed
surface points that remained unchanged between the 1979–1985 and 2004–2009 campaigns were used. At these points, the
elevation differences between the reference DTM and the DSMs were calculated. The reference DTM was the same as the one
used for generating the VHM (see section 2.2.3).
The second dataset comprised 565 selected geodetic control points (i.e., measured with GNSS or triangulation) distributed
throughout Switzerland. This dataset is monitored and maintained by swisstopo; the documented vertical accuracy ranges from
0.003 to 0.1 m, with 42% of the points showing accuracies of 0.05 m or better. For the remaining ones, an accuracy of 0.1 m
is documented (swisstopo, 2023). These geodetic points enabled a completely independent estimation of the vertical accuracy
of the derived DSMs.



As a third reference dataset, 16,650 stereoscopic elevation measurements conducted by a stereo-image interpreter at the Swiss
Federal Institute for Forest, Snow and Landscape Research (WSL) were used. The data collection scheme was the same as in
the NFI (Ginzler, 2019). These measurements were derived only for epoch 1 and within the 666 forest plots of 50 × 50 m area
intersecting the 6 study sites (Fig. 1). For each forest plot, 25 elevation measurements were extracted on the same stereo pair
as used for generating the historical DSM (Fig. A1). In addition, the six land cover classes used to assess completeness were
assigned to each stereo measurement. The 'closed forest' land cover class was further divided into coniferous or deciduous
trees.

**2.3.2 DSM completeness**

Completeness is an essential quality measure for photogrammetric DSM generation. Here, it was computed as the percentage
of successfully matched points out of the potential total number of matched points within a defined spatial unit. Completeness
was calculated within circles with a radius of 5 m centred around points of the land cover statistics dataset. Then, the mean
completeness was calculated for the six land cover classes (i.e., sealed surface, bare land, grass/herb, shrub, closed forest, and
glacial and perpetual snow). Completeness values were calculated in the six study sites for all four epochs.

**2.3.3 Vertical accuracy**

Vertical accuracy was calculated as the difference between the reference dataset (see section 2.3.1) and the DSMs. First, the
elevation differences between the reference DTM and the DSMs were calculated at points classified as sealed surfaces. In
addition, points classified as grass/herb and bare land were used to quantify the elevation bias of the DSMs to terrain slope,
aspect and elevation. Sealed surface points were not used in this investigation because they were mainly located in flat areas.
In a second analysis, the elevation values of the DSMs were compared with approximately 500 independent geodetic points.
Finally, > 16,000 stereoscopic elevation measurements were used to assess the performance of the image-matching workflow
on deciduous and coniferous trees, the primary target objects. It is worth noting that this latter comparison was used to evaluate
agreement rather than serving as a measure of accuracy, since the stereoscopic measurements have some inherent inaccuracies.
To calculate the statistical measures, differences greater than ±50 m between the reference measurements and DSMs were
excluded because such values indicated a failure in image-matching due to cloud cover or saturated images, yielding unrealistic
elevation values. From the remaining differences, robust statistics, such as the median and the normalised median absolute
deviation (NMAD) were calculated, as biases in spatial datasets are often not normally distributed (Höhle and Höhle, 2009).
The NMAD is defined as 1.4826*MAD (mean absolute deviation). Root mean square error (RMSE) values were additionally
calculated, as a standard quality metric to facilitate the comparison of our results with those from other studies. RMSE values
were determined after applying outlier removal (mean ± 3 Std).



## 3 Results

### 3.1 Generation of DSMs

Four countrywide DSMs were generated successfully with the NGATE package of SocetSet. Data voids are related to the lack of stereo images (Fig. 2) or to unsuccessful image-matching in saturated or cloudy areas. Figure 5 shows the hillshaded DSM for the epoch 1979–1985 for the six study sites.

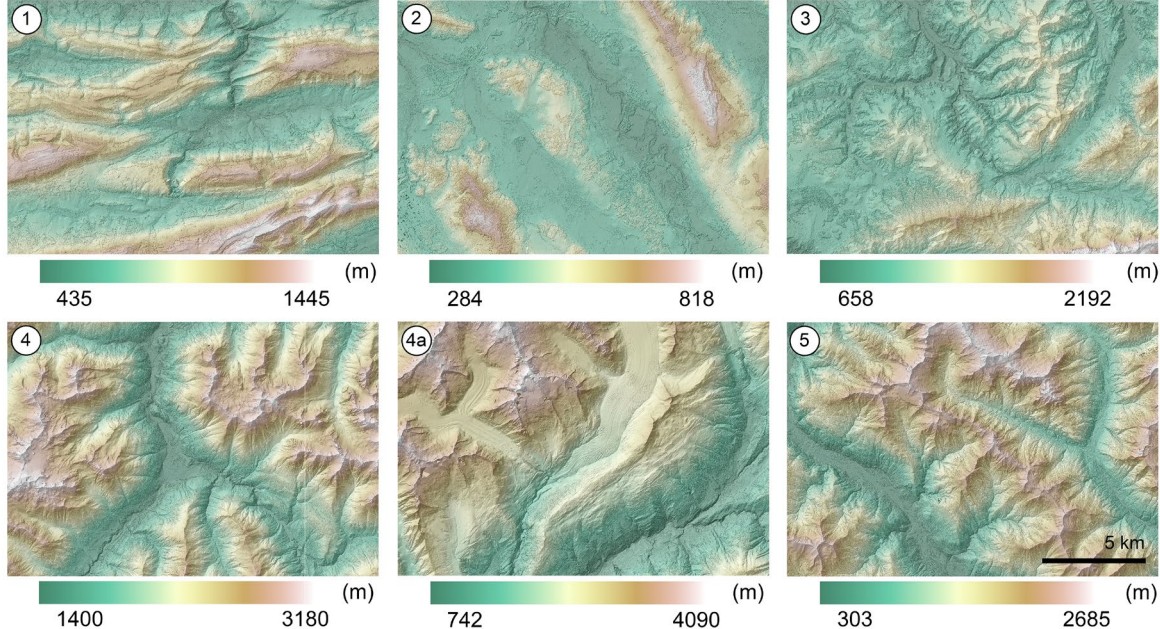

**Figure 5: Hillshaded digital surface models (DSMs) for the six selected study sites and epoch 1 (1979–1985). The year of the DSM is 1982 for sites 1 and 2, 1980 for site 3, 1985 for sites 4 and 4a, and 1983 for site 5.**

### 3.2 DSM completeness

Figure 6 shows the DSM completeness percentages for epoch 1 and the six land cover classes: sealed surface, bare land, grass/herb, shrub, closed forest, and glacial and perpetual snow. The number of sample points for each class is also provided in the figure. Completeness results for epochs 2, 3 and 4 are provided in Appendix A (Fig. A2), and Table A1 shows the corresponding mean values. The lowest completeness occurred in areas classified as glacial and perpetual snow, with a mean completeness of 65% across the four epochs. This result was expected, due to the textureless black-and-white images of these surfaces, which resulted in poor image-matching performance. The highest completeness was achieved for sealed surfaces, with a mean of 98%. For the classes bare land, shrub and closed forest, the mean completeness was approximately 90%, whereas the grass/herb class had a mean of 79%. However, the completeness for the grass/herb class was higher for epochs 2, 3 and 4 (Fig. A2), with a minimum value of 84%.

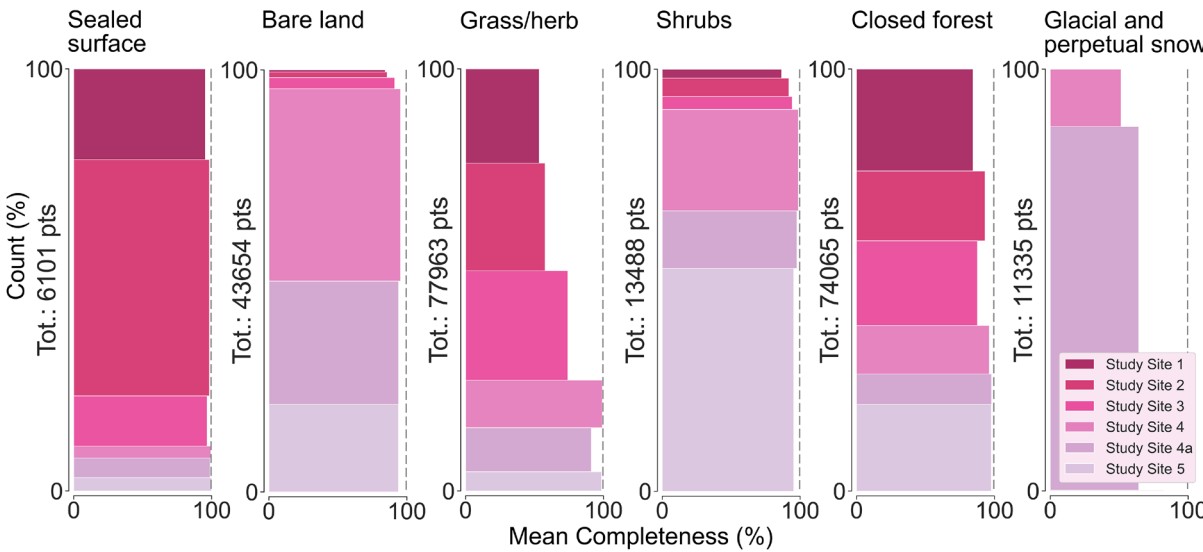

**Figure 6: Distribution of the mean completeness values (as percentages) for epoch 1 across the six study sites and the six land cover classes – sealed surface, bare land, grass/herb, shrub, closed forest, and glacial and perpetual snow. The snow class is only present in study sites 4, 4a and 5. Note that for some classes and study sites, the number of points is minimal. For example, there are only 18 points in study site 5, which is equivalent to 0.15% for the class 'glacial and perpetual snow', and therefore the horizontal bar is not visible.**

### 3.3 DSM vertical accuracy assessment

The vertical accuracy of the DSMs for each epoch was assessed by comparing them with three reference elevation datasets (see section 2.3.1). Figure 7a shows the spatial distributions of the sealed surface points and the reference geodetic points, while Table 2 summarises the statistical measures. Sealed surface points exhibit a clustered distribution in low-elevation areas (i.e., regions 1, 2 and 3) and a sparser distribution in the Alps and Southern Alps (regions 4 and 5, respectively). No clear systematic difference is observed when visualising the positive and negative elevation differences at sealed surface points between the DTM and the DSM from the first epoch. However, plotting the frequency distribution of elevation differences across the entire country reveals a negatively skewed distribution for all epochs (Fig. 7b). Negative differences, where the reference DTM is lower than the historical DSM, may result in the suboptimal location of sealed surface points close to the urban area and forest edge, which could explain the presence of several values < -10 m. The negative skew is also reflected in the median values of the elevation differences (Table 2), with values between -0.3 and -0.5 m and an NMAD of approximately 1.5 m.

In the analysis of the vertical accuracy of the DSMs in comparison to independent geodetic points, the median is around zero, with an NMAD of < 1 m (Table 2). Note that the number of geodetic points is two orders of magnitude smaller than the sealed surface points and more evenly distributed over the country. Additionally, the number of geodetic points varies slightly across the different epochs due to the spatial coverage of the image-matching results (e.g., cloud cover and image saturation). The

286  differences between the geodetic points and DSMs have a more symmetric distribution for all four epochs, with few positive

287  differences > 10 m (Fig. 7c).

288

**Figure 7: (a) Spatial distribution of the sealed surface points and the geodetic points used for the vertical accuracy assessment of the historical digital surface models (DSMs). Sealed surface points are depicted in blue and red, representing the positive and negative differences, respectively, between the reference digital terrain model (DTM) and the DSM for epoch 1. The inset shows the distribution of the sealed surface points and the geodetic points for study site 2, overlaid on the orthophoto for the year 2021 (swisstopo, 2023). Frequency distributions of the elevation difference in metres between the DTM (swissALTI3D) and the DSM (b) on sealed surface points and (c) on geodetic points for the whole of Switzerland for the four epochs. The vertical dashed line indicates the median value. The bin size is 0.5 m.**

296
297
298

**Table 2: Elevation difference between the digital terrain model (DTM) and the historical digital surface models (DSMs) over the sealed surface points and the geodetic points over Switzerland for the four epochs. The number of points before and after outlier filtering is reported; the latter number was used for the root mean square error (RMSE) calculation.**

| Epochs | Accuracy on sealed surface points | | | | | Accuracy on geodetic points | | | | |
|---|---|---|---|---|---|---|---|---|---|---|
| | No. points (±50 m) | Median (m) | NMAD (m) | No. points after filter (± 3 Std) | RMSE (m) | No. points (±50 m) | Median (m) | NMAD (m) | No. points after filtering (± 3 Std) | RMSE (m) |
| Epoch 1 | 90509 | -0.28 | 1.08 | 86993 | 2.98 | 551 | 0.05 | 0.96 | 532 | 2.75 |
| Epoch 2 | 111761 | -0.35 | 1.11 | 107496 | 3.12 | 542 | 0.03 | 0.82 | 526 | 3.25 |
| Epoch 3 | 108358 | -0.41 | 1.14 | 104032 | 3.40 | 536 | 0.02 | 0.78 | 517 | 2.91 |
| Epoch 4 | 102338 | -0.53 | 1.41 | 98157 | 3.90 | 525 | 0.00 | 0.98 | 509 | 3.33 |

The elevation difference between the DTM and the historical DSMs over grass/herb and bare land points shows no significant bias with slope, aspect or elevation, as shown in Figure 8. However, for elevations above 3000 m a.s.l. and slopes greater than 45°, the NMAD increases drastically, exceeding 2 m for all four countrywide DSMs. However, points within these intervals represent only approximately 5% (slope) and 1% (elevation), as shown in the histograms in Fig. 8. The elevation bias dependence on aspect shows a potential shift, with median values below 0.5 m and a constant NMAD of ± 0.8 m for all datasets and aspect intervals.

The agreement between the stereoscopic measurement (see section 2.3.3) taken within the NFI plots and the historical DSM of epoch 1 is reported in Table 3 for the different land cover classes, including a division of the closed forest class into deciduous and coniferous trees. Figure 9 shows one NFI plot with the stereo measurement location and elevation values compared with the DSM profile. As shown in Table 3, the closest agreement is observed for the sealed surface class, with a median difference of 0.19 m and an NMAD of 0.63 m, comparable to the results obtained when analysing the difference between the DTM and DSM for this land cover class (see Table 2). For the grass/herb class, the differences to the stereo measurements exhibit a similar offset but with a larger spread, with a median of -0.15 m and an NMAD of about 1 m. Sample points from the shrub and bare land classes show a wider spread, with NMAD values of 1.62 m and 1 m, respectively, and larger systematic biases, with median values greater than -0.4 m. The weakest agreement is found for the closed forest class, where deciduous and coniferous trees yield comparable results, with median values of about -1.8 m and NMAD values of up to 3.77 m.



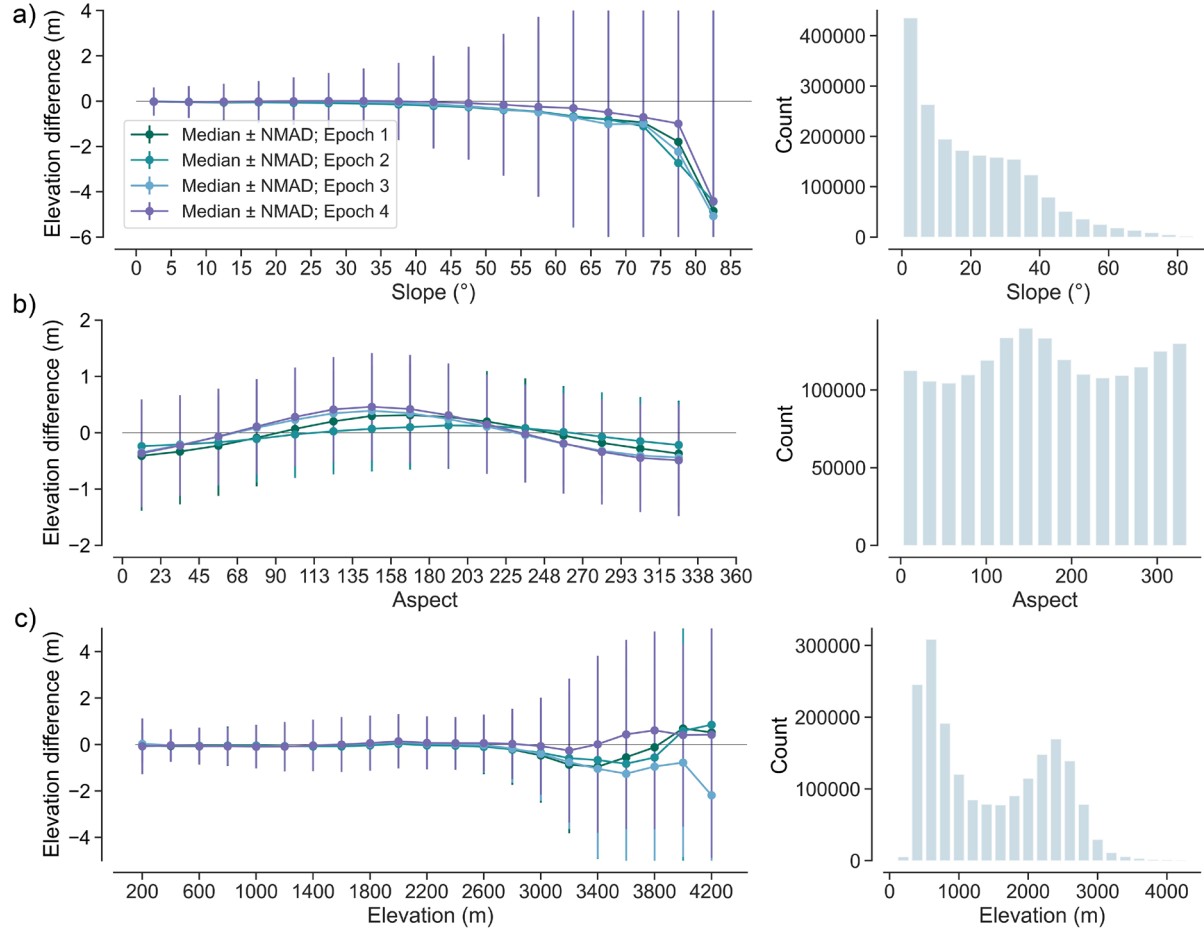

**Figure 8: Elevation differences in grass/herb and bare land points as a function of slope (a), aspect (b), and elevation (c). The histogram on the right shows the number of points for each interval. Elevation differences are calculated as the reference digital terrain model (DTM) minus historical digital surface models (DSMs). Slope, aspect and elevations were derived from the reference DTM.**

**Table 3: Elevation differences over different land cover classes between stereo measurements and the historical digital surface model (DSM) from epoch 1.**

| Land cover class | No. points | Median (m) | NMAD (m) | No. points after filtering (± 3 Std) | RMSE (m) |
|---|---|---|---|---|---|
| Closed forest, deciduous trees | 2917 | -1.83 | 2.75 | 2855 | 4.35 |
| Closed forest, coniferous trees | 2478 | -1.77 | 3.77 | 2441 | 5.47 |
| Grass/herb | 6792 | -0.15 | 1.02 | 6634 | 1.6 |
| Shrub | 424 | -0.41 | 1.62 | 415 | 2.35 |
| Sealed surfaces | 408 | 0.19 | 0.63 | 401 | 1.24 |
| Bare land | 1231 | -0.74 | 1.00 | 1219 | 1.43 |
| Glacial and perpetual snow | 567 | 0.17 | 2.15 | 540 | 2.99 |



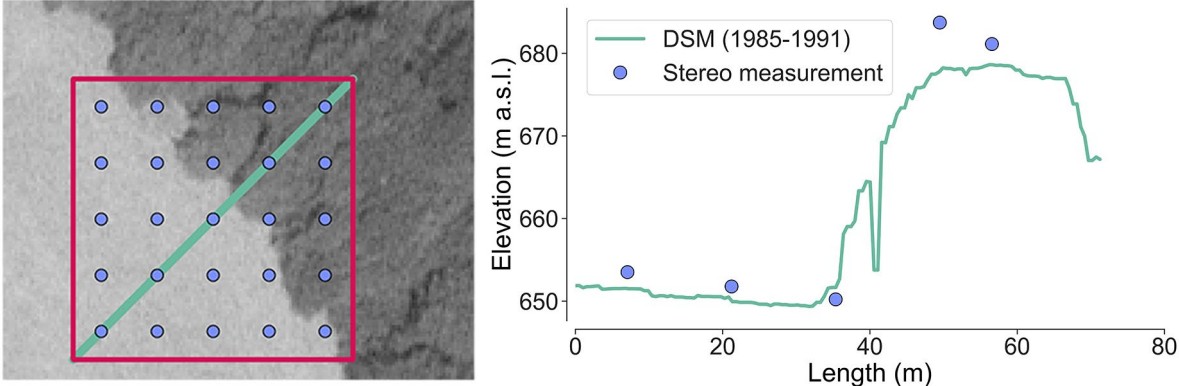

**Figure 9: Stereo measurements within a Swiss National Forest Inventory (NFI) forest plot on the historical orthophoto of epoch 1 (left). On the right, a comparison is depicted between the digital surface model (DSM) profile (green line, interval of 1 m) and the elevation values of the stereo measurements.**

### 3.4 VHMs and forest change

Based on the four countrywide historical DSMs for Switzerland, four VHMs were generated by subtracting a countrywide DTM and applying a non-vegetation mask (see section 2.2.3). Figure 10 shows the VHMs and the forest structure for the six study sites for epoch 2. Since the VHMs are direct derivatives of the DSMs, the statement above about the quality remains valid.

Forest dynamics, such as growth, cutting and regrowth between observation epochs are evident over the six study sites, as shown by the height differences of the VHMs between consecutive epochs (Fig. A3). A detailed look at the VHM profiles of study sites 2 and 4 (Fig. 10b) shows minimal change in forest height in mature forest stands (VHM > 20 m), indicating consistency among the VHMs. However, forest growth, cuts or damage followed by regrowth can be detected, for example, between epoch 4 and 2022 (Fig. 10). The Swiss Plateau forest production region, where study site 2 is located, was heavily affected by storm damage. Figure 11 shows a small area of the VHM in region 2 for epochs 1, 3 and 4, overlaid by polygons indicating areas damaged by the two largest storm events recorded in Switzerland: Vivian (25–27 February 1990) and Lothar (26 December 1999). The VHMs effectively capture the storm-affected areas. The VHM of 1982 depicts forest cover in all storm-affected areas, while the 1994 VHM clearly shows the damage caused by storm Vivian. The areas impacted by storm Lothar show mature forest stands in the 1994 VHM, while the heavily damaged areas are visible in the 2002 VHM.

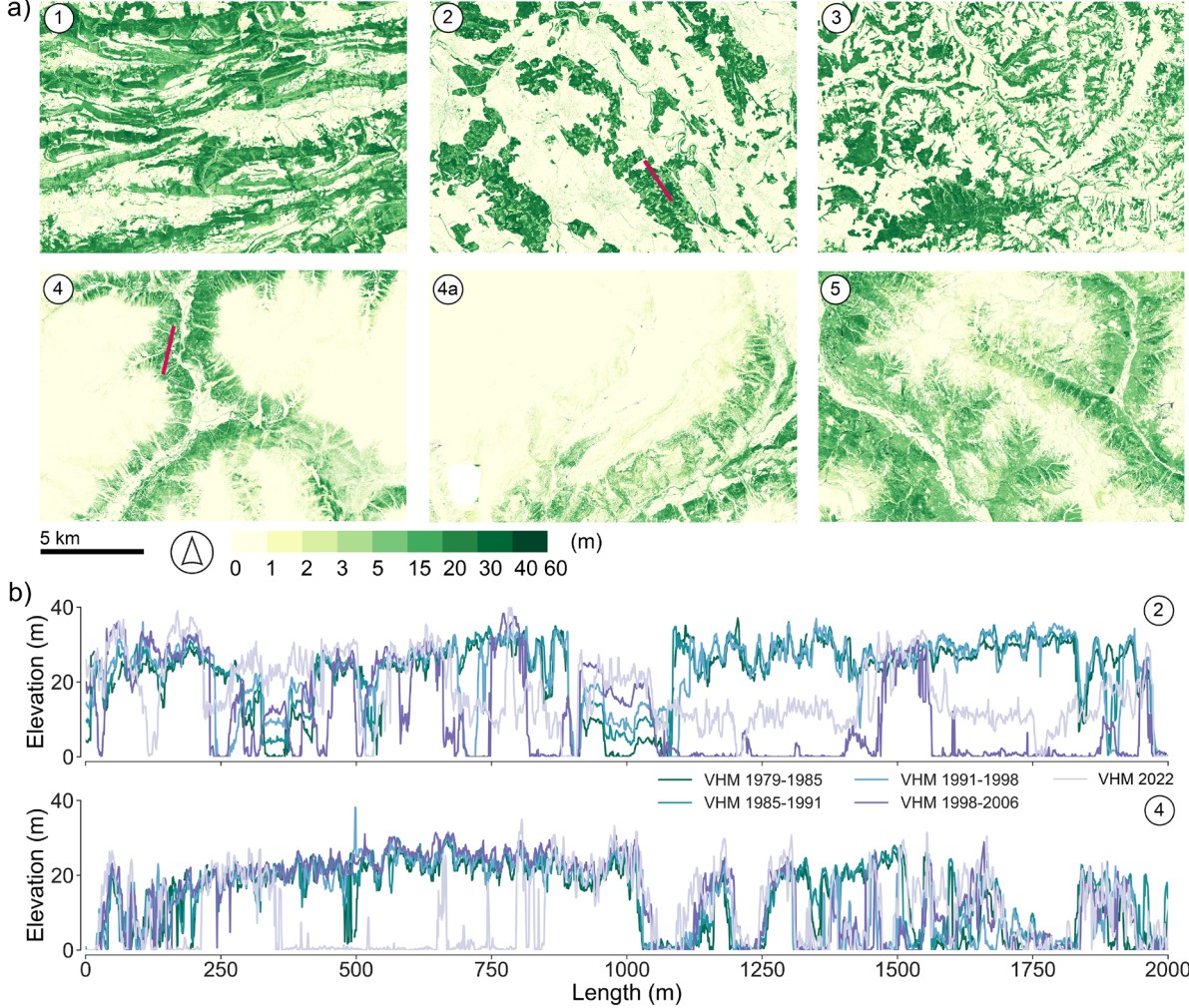

**Figure 10: Illustration of (a) the vegetation height models (VHMs) of epoch 2 (1985–1991) for the six selected study sites. The VHMs correspond to the years: 1987 for site 1, 1988 for site 2, 1986 for sites 3 and 4a, 1991 for sites 4, and 1989 for site 5. (b) The profile of the VHMs from historical aerial images for the four epochs and from a modern airborne survey for study sites 2 (acquisition period 2018 and 2022) and 4 (acquisition period 2022). The 2 km profile line is shown in red in (a).**

Besides these qualitative results, these historical VHMs have been used in various studies (Price et al., 2020; Ginzler et al., 2021). Ginzler et al. (2021) investigated forest dynamics in Switzerland between 1980 and 2010. Their analysis was based on aggregated average vegetation heights for 100 × 100 m cells derived from the VHMs. They quantified forest dynamics by retrieving the absolute height change over the entire period (sum of absolute differences) and the classified height change between periods (height gain or loss). The results showed high forest dynamics in the Swiss Plateau and some pre-alpine areas in the northern Alps, while significant parts of the Jura and southern Switzerland showed little change during these 30 years. Price et al. (2020) used the historical VHMs to map and monitor woody above-ground biomass (AGB) dynamics across

Switzerland over 35 years (1983–2017). They found a consistent relationship between vegetation height derived from the
VHMs and NFI measures of woody AGB across four inventory periods.

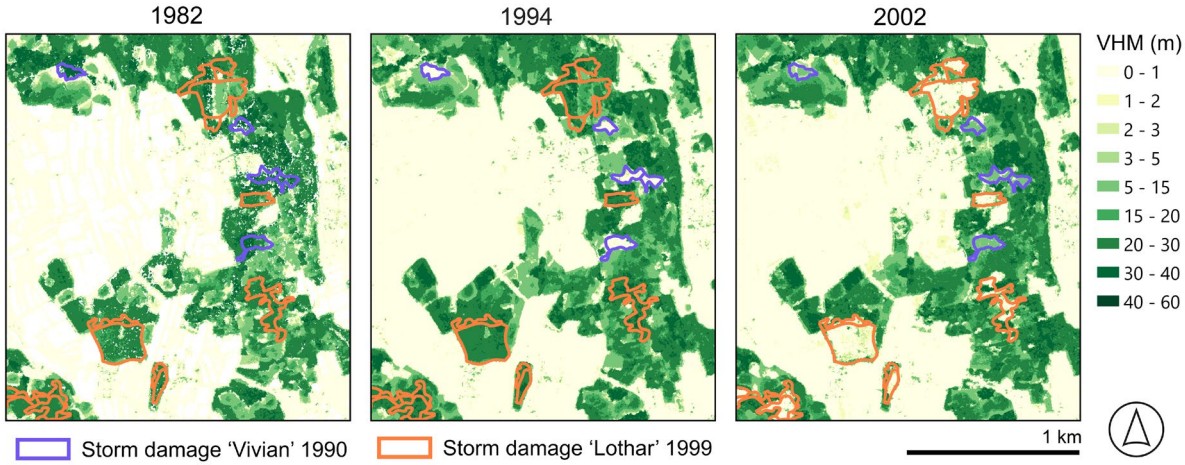


**Figure 11: Small extent of region 2 (Swiss Plateau), showing the vegetation height models (VHMs) of epochs 1, 2 and 4, overlaid by**
**polygons indicating areas heavily affected by two major storm events in Switzerland (Vivian: 25 to 27.02.1990 and Lothar:**
**26.12.1999).**
**4 Discussion**
**4.1 Potential of countrywide historical DSMs and VHMs of Switzerland**
This study underscores the great value of historical aerial images for reconstructing accurate and high-resolution DSMs over
approximately three decades. For the epochs 1979–1985, 1985–1991, 1991–1998 and 1998–2006, we successfully processed
four countrywide DSMs and VHMs at a resolution of 1 m. The quality of the generated models is sufficient for various
geoscience applications, such as the quantification of natural hazards, mass movements, and sediment erosion, and the
monitoring of human interference (e.g., deforestation) and urban development. Our historical VHM products show a high level
of consistency, suggesting that changes in forest height are real and significant and thus are valuable tools for understanding
forest dynamics in Switzerland. This includes the tracking of forest growth and changes in forest cover over time, the
differentiation between disturbed (due to management practices or storm events) and undisturbed forests, and the detection of
regrowth after disturbance. The clear delineation of storm-damaged areas (e.g., from storms Vivian and Lothar) in our VHM
data demonstrates the capability of the models to identify disturbances accurately. By analysing the temporal changes in these
areas, we can differentiate between disturbances caused by natural events and those resulting from forest management
activities. Therefore, insight can be gained into post-storm regrowth and forest management activities across the forest
production regions, and their effectiveness can be quantified, for example in terms of protection forest function. This dataset
offers great added value to the Swiss NFI forest management data and supports future management strategies.



## 4.2 Quality of the DSMs

The completeness of the image-matching shows a steady increase over the four epochs (Fig. A2), particularly in areas classified as grasses and herbs. This improvement is strongly related to advancements in camera technology (models and lenses), image-matching results, and the quality of the exterior orientation provided in this study.

In areas with successful image-matching, countrywide accuracy analysis comparing derived DSM elevations to swissALTI3D DTM elevations at sealed surface points shows metric to sub-metric accuracy with an uncertainty (NMAD) < 2 m. However, a comparison with geodetic points distributed across Switzerland shows even higher, decametric accuracy.

The negative bias, where reference data was lower than the historical DSMs, may have resulted from the suboptimal location of sealed surface points near urban areas and forest edges (Fig. 12a), which could explain the presence of several values more negative than -10 m (Fig. 7b). The comparison with geodetic points shows more positive outliers with values above 10 m (Fig. 7c), likely related to the location of reference geodetic points on objects such as rooftops and statues, which cannot be precisely reconstructed from historical photographs (Fig. 12b).

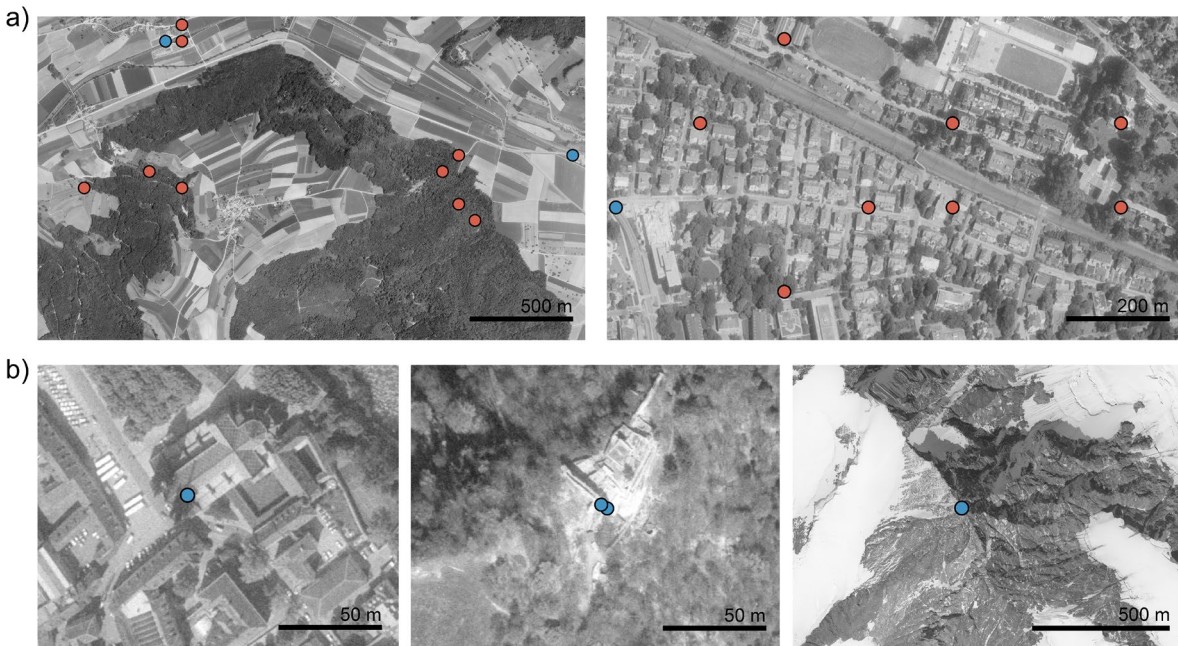

**Figure 12: Example of suboptimal locations of (a) sealed surface points and (b) geodetic points used for the accuracy assessment. Positive biases over these points are shown in blue, whilst negative ones are shown in red.**

An elevation bias in areas with slopes greater than 45° and elevations above 3000 m a.s.l. (Fig. 8a, c) is present in many photogrammetric products from airborne and satellite sources (e.g., Piermattei et al., 2019), and is more pronounced in historical images with 8-bit radiometric resolution and lower accuracy in exterior orientation. Steep slopes and mountainous areas often experience strong shadows, overexposed photographs on snow-covered surfaces, and clouds over mountain tops, all affecting image-matching results and vertical accuracy, and increasing noise in the DSMs.





Comparing the agreement between manual stereo measurements and the historical DSM of epoch 1, the closed forest land
cover class shows the largest bias among all classes. However, similar patterns are visible when comparing manual stereo
measurements with a DSM from images acquired by a modern near-infrared Leica ADS Sensor with a GSD of 0.5 m (Ginzler
et al., 2015). Ginzler et al. (2015) achieved median values of -0.34 m (coniferous) and -0.79 m (deciduous), with NMAD
values in the same range as in this study. The larger bias in historical data might be due to the more demanding visual detection
of objects during manual stereo measurements on panchromatic imagery and the coarser resolution of the images. This
observation is supported by Ginzler et al. (2015), where stereo measurements on imagery with 0.25 m GSD reduced median
values to -0.08 m (coniferous) and 0.16 m (deciduous). All other land cover classes show median values between ~0.1 m and
0.7 m and NMAD values between ~1 m and 1.5 m in both studies.

## 4.3 Challenges of image processing and limitations of our dataset

Challenges in working with airborne historical aerial images can include difficulties accessing the archive, the high cost of
images, the low quality of scanning film negatives, and the poor availability of metadata such as exterior and interior
orientation. In Switzerland, swisstopo provides free access to a comprehensive archive of scanned historical aerial images
covering the entire country, along with calibration protocols and exterior orientation parameters for each image, essential for
photogrammetric processing. However, the orientation accuracy of historical images varies between epochs, requiring a
vertical correction of the DSMs due to offsets of several metres. Often, historical images lack calibration protocols and exterior
orientation information, requiring ground control points to calculate the camera location. This has led the scientific community
to adopt 'structure from motion' photogrammetric approaches for processing historical aerial images (e.g., Muhammed et al.,
2023), as these approaches do not require approximated values, such as camera calibration information. Additionally, pipelines
have been developed to minimise the effort of collecting GCPs (e.g., Knuth et al., 2023).
To handle a countrywide dataset acquired over three decades, the processing of approximately 40,000 images, as well as post-
processing including the vertical correction of about 20,000 DSMs, was done at the map sheet level. Vertical correction was
required to minimise offsets caused by an inaccurate exterior orientation of the images (see section 2.2.2). Each corrected map
sheet DSM was merged without further consideration of the edges of adjacent map sheets, potentially creating small jumps at
the edges. These jumps are negligible at the country scale, but further co-registration should be considered for local and
catchment studies involving more than one map sheet.
Another challenge of historical black-and-white imagery with 8-bit radiometric resolution is oversaturation in snow-covered
or heavily illuminated areas, leading to no or erroneous image-matching results, particularly in mountainous regions. This is
evident in the completeness results for points classified as glacial and perpetual snow (see section 3.2). Therefore, for studies
related to glaciers and snow, the noise and the voids in the generated DSMs must be addressed with proper void filling and
filtering. We used a simple TIN interpolation to fill the voids, providing a binary mask indicating interpolated areas, which
can be used to remove these areas for further analysis. For glaciological studies or applications requiring precise dates, we do
not have the exact information on the dates of each DSM/pixel, but we provide the year of the survey campaign per map sheet.



## 5 Data availability

Datasets can be accessed from EnviDat (https://doi.org/10.16904/envidat.528, Marty et al., 2024). The following files are available for the four epochs: countrywide digital surface model (DSM), hillshaded DSM, and vegetation height models (VHMs). A metadata shapefile is provided with information about the acquisition year of the photographs used here; the geometry corresponds to the 1:25,000 map sheets used during this study. In addition, a binary raster is provided indicating successfully matched pixels (1) and interpolated pixels (0), and a non-vegetation mask raster 'TLM_Vegetation_mask' is provided, indicating vegetated areas (1) and non-vegetated areas (0). All the raster files are provided as GeoTIFF files.

## 6 Conclusions

Millions of aerial photographs dating from the First World War to the early 2000s represent an enormous, yet largely untapped, resource for geoscientists, with significant potential for documenting and quantifying surface changes over the last century. In this study, we used freely available historical stereo images covering the whole of Switzerland, processed using photogrammetric software. These images were provided with both exterior and interior orientation, allowing us to derive four countrywide DSMs at a 1 m spatial resolution across four epochs, with survey campaigns spanning approximately seven years. Our developed workflow achieved sub-metric accuracy and high completeness of the DSMs in nearly all regions of Switzerland, demonstrating the feasibility of capturing continuous surface change patterns at a high spatial resolution over different land cover classes. However, lower completeness and increased noise were present in snow-covered areas, due to image saturation and reduced performance of the image-matching algorithm in these areas. As our study was conducted as part of the Swiss NFI, we also generated four countrywide VHMs using an available countrywide DTM. These VHMs have been used to quantify forest changes (Ginzler et al., 2021) and monitor above-ground biomass (Price et al., 2020). The high consistency of the VHMs indicates their potential for future research into forest ecosystem dynamics, services and management.

Future work can build upon these datasets by integrating modern data to extend the observation period, thereby enhancing the detection of climate signals. Additionally, refining time series methods for very-high-resolution datasets will further improve our understanding of surface changes and related processes.

Finally, this study highlights the importance of recognising the value of historical aerial imagery datasets. We urge mapping agencies, governmental authorities and military services to unlock these archives and make them freely accessible online. The creation of countrywide historical aerial image DSMs is feasible only when such data is readily available, as demonstrated by the open access provided by swisstopo in Switzerland.





## Appendix A

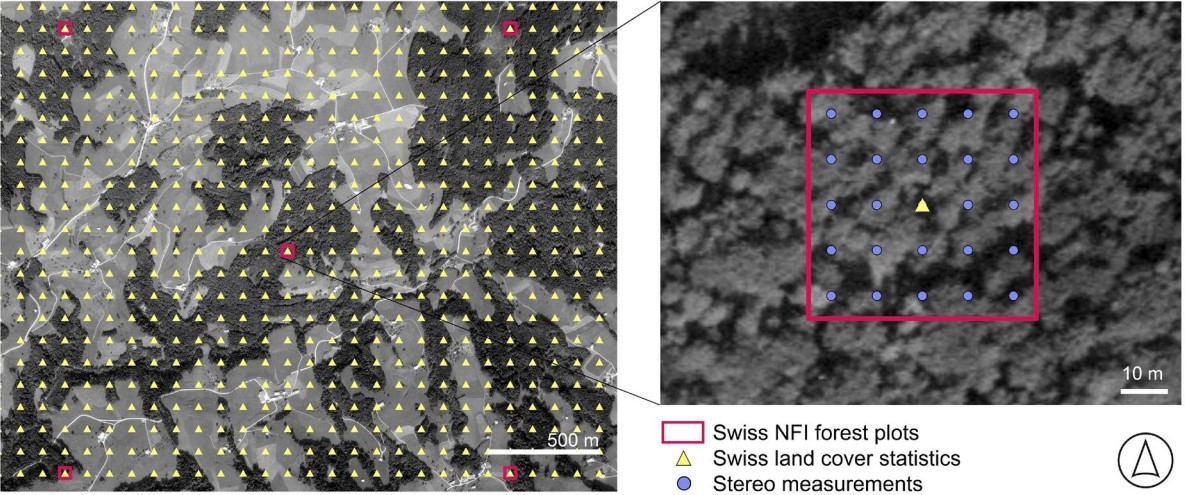

**Figure A1: Illustration of the distribution of the Swiss land cover statistic points and the Swiss National Forest Inventory (NFI) forest plots with the stereo measurements. The background is the generated orthophoto from the first epoch (1979–1985).**







Figure A2: Distribution of the mean completeness values, as percentages, for (a) epoch 2, (b) epoch 3, and (c) epoch 4, across the six study sites and the six land cover classes – sealed surface, bare land, grass/herb, shrub, closed forest, and glacial and perpetual snow. The last class is only present in study sites 4, 4a and 5.





**Table A1: Mean completeness values (as percentages) for each epoch over the six study sites, and for the six land cover classes. For**
**each class, the number of points used to assess the accuracy is provided. n.a. indicates not available.**

| Study site | Sealed surface | | Bare land | | Grass/herb | | Shrub | | Closed forest | | Glacial and perpetual snow | |
|---|---|---|---|---|---|---|---|---|---|---|---|---|
| | Mean (%) | Count (No.) | Mean (%) | Count (No.) | Mean (%) | Count (No.) | Mean (%) | Count (No.) | Mean (%) | Count (No.) | Mean (%) | Count (No.) |
| **Epoch 1** | | | | | | | | | | | | |
| 1 | 95 | 1312 | 84 | 221 | 53 | 17428 | 87 | 286 | 84 | 17899 | n.a. | n.a. |
| 2 | 98 | 3416 | 86 | 565 | 58 | 19879 | 92 | 586 | 93 | 12254 | n.a. | n.a. |
| 3 | 97 | 726 | 91 | 1172 | 74 | 20202 | 94 | 409 | 88 | 14877 | n.a. | n.a. |
| 4 | 99 | 173 | 95 | 19892 | 99 | 8785 | 99 | 3244 | 96 | 8496 | 51 | 1543 |
| 4a | 99 | 284 | 94 | 12759 | 91 | 8119 | 98 | 1838 | 98 | 5346 | 64 | 9774 |
| 5 | 100 | 190 | 94 | 9045 | 99 | 3550 | 95 | 7125 | 98 | 15193 | 82 | 18 |
| **Epoch 2** | | | | | | | | | | | | |
| 1 | 97 | 1526 | 90 | 287 | 84 | 21520 | 93 | 331 | 92 | 22974 | n.a. | n.a. |
| 2 | 99 | 3669 | 97 | 606 | 93 | 22699 | 98 | 646 | 97 | 13934 | n.a. | n.a. |
| 3 | 98 | 776 | 93 | 956 | 94 | 21609 | 96 | 434 | 88 | 16075 | n.a. | n.a. |
| 4 | 99 | 176 | 94 | 17811 | 96 | 8638 | 98 | 3139 | 95 | 8477 | 67 | 951 |
| 4a | 100 | 296 | 92 | 11497 | 99 | 7163 | 98 | 1593 | 95 | 5287 | 78 | 9155 |
| 5 | 100 | 198 | 93 | 9699 | 99 | 3768 | 96 | 7330 | 97 | 16393 | 76 | 22 |
| **Epoch 3** | | | | | | | | | | | | |
| 1 | 94 | 1947 | 87 | 342 | 91 | 24952 | 96 | 408 | 91 | 25816 | n.a. | n.a. |
| 2 | 99 | 3718 | 95 | 618 | 92 | 23317 | 95 | 639 | 92 | 13841 | n.a. | n.a. |
| 3 | 98 | 819 | 91 | 1400 | 95 | 23259 | 88 | 514 | 94 | 17229 | n.a. | n.a. |
| 4 | 99 | 163 | 93 | 17761 | 96 | 8437 | 98 | 3158 | 96 | 8239 | 68 | 1312 |
| 4a | 99 | 360 | 94 | 13907 | 98 | 8868 | 98 | 1964 | 97 | 6060 | 85 | 12399 |
| 5 | 100 | 233 | 87 | 10475 | 97 | 4106 | 88 | 8122 | 94 | 17776 | 87 | 22 |
| **Epoch 4** | | | | | | | | | | | | |
| 1 | 99 | 1748 | 96 | 281 | 89 | 22408 | 99 | 348 | 98 | 22718 | n.a. | n.a. |
| 2 | 99 | 3784 | 97 | 560 | 86 | 21622 | 96 | 670 | 94 | 15078 | n.a. | n.a. |
| 3 | 98 | 840 | 94 | 1715 | 95 | 23288 | 97 | 553 | 94 | 17637 | n.a. | n.a. |
| 4 | 99 | 147 | 93 | 15428 | 98 | 7680 | 96 | 3008 | 92 | 7927 | 85 | 953 |
| 4a | 99 | 365 | 90 | 12840 | 97 | 8811 | 96 | 1862 | 96 | 5623 | 85 | 11283 |
| 5 | 99 | 154 | 81 | 4235 | 93 | 2175 | 84 | 4497 | 88 | 11808 | 27 | 5 |




**Figure A3: The height difference between the vegetation height models (VHM) of consecutive epochs for each study site (1–5). The black line in study sites 2 and 4 indicates the location of the profile shown in Figure 10.**




**Supplement**
No supplemental material.
**Author Contributions**
MM and CG designed the study. MM processed and generated all the data. LP and MM analysed the dataset. LP prepared all
the figures and tables. LP and MM wrote the manuscript. All authors contributed to the discussion and editing of the text.
**Competing interests**
The authors declare that they have no conflicts of interest.
**Acknowledgements**
This study was supported by the Swiss National Forest Inventory (NFI). We thank the Federal Statistical Office FSO and the
Federal Office of Topography swisstopo for providing access to the historical photographs. We are grateful to Holger Heisig
(swisstopo) for the successful semi-automatic orientation of thousands of historical images and fruitful and interesting
discussions about this source of data and to Dr Melissa Dawes for professional language editing.
**Funding**
This study was carried out in the framework of the Swiss National Forest Inventory (NFI), a cooperative effort between the
Swiss Federal Institute for Forest, Snow and Landscape Research (WSL) and the Swiss Federal Office for the Environment
(FOEN).

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
