# Peer review of "Countrywide Digital Surface Models and Vegetation Height Models"

_Earth System Science Data, 2024_

## Author Comment (AC1)

Title: **Countrywide Digital Surface Models and Vegetation Height Models from Historical Aerial Images**

Author(s): Mauro Marty et al.

Handling editor: Alexander Gruber, alexander.gruber@geo.tuwien.ac.at

**Author's response related to the comments received on the manuscript:**
**https://doi.org/10.5194/essd-2024-428**

We thank the two reviewers for their feedback on our manuscript. We revised our manuscript after considering their comments and suggestions, which helped to improve the clarity and correctness of our paper.

The main changes we made to the manuscript are as follows:
- We recalculate the statistics using the difference between the historical DSMs and the reference DTM, rather than vice versa. We also apply outlier filtering through robust statistical methods.
- To enhance the vertical accuracy analysis, we exclude sealed surface points near buildings and forests, as these can skew the statistics.
- Furthermore, we have improved our description of the image processing and DSM production processes. We now provide more detailed information regarding image orientation and clarify how the obtained accuracy relates to pixel resolution.

We are convinced that these revisions, as detailed in the point-by-point reply below, address all the comments raised by reviewers and significantly improves the quality of our study.

**Point-by-point reply to reviewer comments**: Reviewers' comments are in black, and the authors' response is in blue, with citations from the revised manuscript in green.

Interesting work although it would benefit to have some more technical details to boost understanding and replicability in other areas, such as:

Response: Thank you very much for reviewing our manuscript and providing constructive comments. We have addressed them as outlined below.

How were the DSM generated in the various epochs? I mean, not sw-wise...for the epoch 1979–1985, did you generate DSM for each year and then average it to have a DSM for the epoch?

Response: Thank you for this comment. We improved the description of the process for the individual epochs in section 2.2.2. In summary, the DSM processing of one epoch follows these steps: first, we perform image matching on the single stereo footprints belonging to that epoch (section 2.2.1). In a second step, we merge all single stereo footprints from the same epoch into 1:25,000 map sheet units and cut them to their borders (see sections 2.2.2 and 2.2.3). To derive the nationwide coverage per epoch, the map sheet DSMs belonging to the same epoch were merged. Since they were cut at the map sheet border, there is no overlap between adjacent map sheets.
R 184-192: "The 20,346 DSMs resulting from image-matching of the single stereo footprints were merged to the level of the 1:25,000 map sheets, which were already used to guide the image acquisition campaigns (see section 2.1). For each map sheet geometry, all DSMs from the same epoch that corresponded to individual stereo footprints were selected based on spatial intersection. They were then merged and cropped to fit the extent of the map sheet, which typically included around 50 stereo footprints in flat areas and up to 150 in high-alpine regions. Since adjacent stereo footprints spatially overlapped, the pixels in these overlapping areas often included multiple elevation values. To address this, the elevation values were merged using the median height. Grid cells without any successfully matched elevation information were interpolated using a triangular irregular network (TIN) surface."
R 231-233: "After correcting for randomly distributed elevation biases in each map sheet DSM and epoch (as described in section 2.2.2), all the corrected map sheet DSMs from the same epoch were mosaiced to a single DSM as a Geotiff file, covering the whole of Switzerland."

In the photogrammetric processing, how did you handle the information of the fiducial marks?

Response: Thank you for the question. We clarified this in the revised text.
R 139-143: "With known interior and exterior orientation parameters, the historical images were imported to the software through the "frame import dialogue". The import process was separately done for all images originating from the same camera. During the frame import process, camera calibration values were entered, and fiducial marks were primarily detected using automatic fiducial detection, implemented in the frame import workflow. Where this

automatic approach failed, manual measurement of fiducial marks was conducted. Subsequently, the DSMs…"

How did you produce the VHM from the available DSM? how was the filtering done? did you use an available DTM? if so, which one?

Response: The paragraphs related to the VHM may have been overlooked. The information on the VHM generation, filtering, and DTM has already been reported in the original text, section 2.2.3. See text below.
R 226: "The four countrywide VHMs were derived by calculating the difference between the historical DSMs and the reference DTM (swissALTI3D version 2017, swisstopo; see workflow in Fig. 3)..."
R 233: "nDSM raster cells with values > 60 m were considered outliers due to erroneous image-matching and thus were replaced by 'no data'..."

How did you assess the quality of the historical DSMs? You mention "evaluating the vertical accuracy and the completeness of image-matching" but what was the ground truth? The reported % are very optimistic

Response: Thank you for this observation. The whole section "2.3 Quality assessment of the DSMs", starting at line 249, deals with a description of the quality assessment of the generated DSMs. We provide a summary below; however, we consider the text on completeness and accuracy assessment to be quite detailed. Minor changes to the original text are made to address the comments of Reviewer 2.

The completeness (section 2.3.2) of the image matching process was assessed by calculating the percentage of successfully matched points within a circle of 5 m around points of the land cover statistics dataset. Assessed land cover types are "Sealed Surface", "Bare land", "Grass/herb", "Shrubs", "Closed forest" and "Glacial and perpetual snow". For each of the study sites, the average completeness per land cover type is documented in Figure 6. Those percentage values are obtained if the Figure of Merit Values are analysed from the resulting DSMs.

Multiple analyses were conducted to assess the vertical accuracy of the DSMs (section 2.3.3). In a first step the difference between the historical DSMs and the reference DTM (ground truth) was analysed at locations of points classified as sealed surface (from the land cover statistics dataset) and to check for dependency of accuracy on terrain slope, aspect and elevation the same was done on points classified as grass/herb. In a second step, DSM heights were compared at approximately 500 locations with independent geodetic points (ground truth). A third analysis compared absolute DSM height with more than 16,000 stereoscopic elevation measurements (ground truth) to primarily investigate the difference in matching performance between deciduous and coniferous trees.

This paper is about the usage of historic aerial images of entire Swiss country to derive digital surface models (DSMs) which are then used to produce vegetation height models. Four epochs are considered ranging from 1979 to 2006. The images are used with existing orientation data. The focus of the paper lies on the accuracy analysis of the derived DSMs using a reference DTM, geodetic check points and manual stereoscopic measurements.

**General comments:**

The paper is well organized and well written. The topic is interesting, showing what problems and which accuracies one might expect when using historic aerial images to derive 3D information. All in all the content is good. However, a few details are missing from my point of view. I raise concerns regarding the sign of differences and, in particular, the statistical analysis (although intended to be robust), which is not done in an optimal way (see major detailed comments). Also a clear statement regarding the relation between the obtained accuracies and the GSD of the images should be given. Therefore the major revision.

Response: Thank you for your very valuable and thorough review of our manuscript. We appreciate the constructive comments and have addressed all major and minor points.

**Major detailed comments:**

Sign of differences

On row 196 you say: "The elevation differences were calculated consistently as the reference data minus the historical DSMs."
--> If there is no meaning attached to A or B, then the differences "A - B" and "B - A" are equally good. However, if one of A and B is a "reference", then the reference should always be subtracted, because the reference defines where the "origin" should lie. Thus if X is the data and the reference is X0 the difference should be computed as X - X0. So, if the reference is zero, you end up getting your original data X - X0 = X - 0 = X and not -X (as in your case).
In your case the reference is a terrain model; thus even more motivating to compute the differences as: historic_DSM minus reference_DTM. Thus your differences can easily be compared with potential object heights (causing these differences), because the signs fit. And the sign of a non-zero offset (mean or median) is easily understandable; thus in a vegetated area a positive offset can be expected.
Luckily the effect on your paper is not big: In your statistics RMS, NMAD and STD will not change; only the mean and median will change. The color codings will only swap their colors.
 Still I think it is crucial that the differences are correctly computed as X - X0.

Response: Thank you for your interesting observation. The choice of performing "reference-generated" (i.e., DTM2017-DSMhistorical) was related to the year of the reference DTM, which is more recent than the historical DSMs. Therefore, any positive changes indicate

that the DTM is higher than the historical one, which is more of an approach adopted for assessing elevation changes. Our comparison for accuracy assessment is also performed on sealed surface points (i.e., stable points), so here the DSM and DTM heights should be identical, and the reference DTM on those points does not have zero elevation. However, since we are comparing a DTM with a historical DSM, which may have a higher elevation in stable points due to potentially vegetation, etc., we agreed with the proposed solution of subtracting the DTM from the DSM (i.e., "generated-reference"). We apply this solution to all analyses. We corrected the statistics in the text (section 2.3.1), tables, and the plots are inverted correspondingly. Note that the plots show the distribution before filtering. We also notice an error in the previous plot as values bigger than 10 were not reported as a grey colour bar, while they are included in the new plots.

Furthermore, we decided to remove sealed points with a distance smaller than 10 m to houses and forests, before calculating the NMAD and median. So Table 2 reports the new median and NMAD values as well as the number of considered sealed surface points. Also, the histogram in Fig. 7b is related to the updated sample of sealed points.

R 282-285: "Vertical accuracy was calculated as the difference between the DSMs and the reference dataset (see section 2.3.1). First, the elevation differences between the DSMs and the reference DTM were calculated at points classified as sealed surfaces.Sealed surface points closer than 10 m to historical and current houses and the forest boundary were removed."

Robust statistics

It is good that you consider the possibility of outliers and that you want to remove them and their influence. However, the current approach is suboptimal. In order to remove outliers from your data you must know the distribution (meaning its type and the values of the defining parameters). In many cases the normal distribution is a good assumption for the type.

If you would know in beforehand the mean and std of the normal distribution behind your corrupted data, then the outliers could be filtered out by [mean +- 3*std] (because > 99.7% of the distribution would be inside this interval).

However, if you do not know the parameter values (i.e. mean and std), then it is a bad idea to compute mean and std from the data that are corrupted by outliers, because mean and std would be computed totally wrong. Instead, these parameters need to be computed by robust methods from these corrupted data. Under the assumption of normal distribution the median is a robust estimate for the mean, and the NMAD is a robust estimate for the std. Consequently, outliers from these corrupted data should be detected by considering the interval [median +- 3*NMAD]. Often, the validity of (a single!) normal distribution is not fully met, then a larger interval should be chosen e.g. +- 5*NMAD.

Response: Thank you for this valuable comment. We fully agree that robust statistics should be used for outlier filtering. We used the suggested 5*NMAD and recalculated the statistics after applying this outlier filter; the new values are reported in the tables and adjusted in the text.

Remark: Your statement on row 242 is not correct: "The NMAD is defined as 1.4826*MAD (mean absolute deviation)." The MAD is the median of absolute deviations. And these deviations are computed with respect to the median of the data. There is no "mean" used at all when computing the NMAD, since you are confronted with outliers. https://en.wikipedia.org/wiki/Median_absolute_deviation

Response: Thank you for spotting this mistake in the text. We corrected it. Our calculation was performed correctly, but it was wrongly reported in the text.
R 296: "The NMAD is defined as 1.4826*MAD (median absolute deviation)."

There are several locations in your paper where you apply the outlier removal wrong by considering the interval [mean +- 3*std]; e.g. in Tab. 2 and Fig 7: For epoch 4 (geodetic points) you obtain median = 0.00 and NMAD = 0.98 m. If the assumption of normal distribution is correct, then the RMS of this distribution should also be close to 1 m. However, >>after<< outlier removal you obtain RMS = 3.33 m. The histogram in Fig 7 (bottom, right) clearly shows that almost all data is within the interval +- 5 m. Thus an outlier removal based on the interval [median +- 5*NMAD] would do the trick. You, however, picked [mean +- 3*std] where the (unreported) std from the corrupted data will be somewhat in the range of the reported RMS of 3.33 m and consequently you accept everything on the interval of roughly +-10m. Another example are the sealed surfaces in Tab. 3, where median = 0.19 m, NMAD = 0.63 m and RMS (after outlier removal) = 1.24 m also do not fit together since under the normal distribution assumption the RMS should be much closer to sqrt(median^2 + NMAD^2).

Response: Very good observation, thank you. We improved our filtering by following the suggestion of using a more robust outlier filtering method (median ± 5*NMAD). We corrected the new estimates of RMSE in Tables 2 and 3, which are now closer to the NMAD. The median and NMAD in the tables are related to the unfiltered values, while RMSE is calculated after filtering. For sealed points, we removed all the points with a distance smaller than 10 m from houses and forest; therefore, the median, NMAD and number of points (before the filtering) are also changing.
The distribution of the differences (Figure 7) displays the unfiltered data, along with the corresponding median, as reported in the table. The histogram in Fig. 7b shows the distribution excluding sealed points close to houses and forests, and as visible, fewer outliers (> 10 m). We ran a Shapiro-Wilk test to check the distribution of the elevation differences on sealed points, as well as the geodetic one (Figure 7b,c, Table 2). We found that they are not normally distributed,

as confirmed by the QQPlot. Therefore, as the residuals are not Gaussian and are tailed and skewed, especially for the sealed points, we added this observation to the manuscript and clarified that the RMSE accuracy metric, which assumes normality, can be misleading.

My last remark on this behalf concerns the vertical correction on page 9, row 167:
"Outliers were filtered out based on mean ± 1 standard deviation". From my outline above it should become obvious why this is a very odd method for outlier removal. The choice of "+- 1 std" is statistically unjustified (since this interval corresponds to only 68%). The reason why you chose it, is quite likely due to the outliers in your data which yield a wrong large std that alone (i.e. 1*std instead of 3*std) is sufficient for your task. It is further odd, that you compute the mean >>before<< outlier removal (and use it for this), but afterwards you compute the "median of the remaining elevation differences" (row 168). In case of robust statistics it should always be the other way: First compute the median and afterwards compute the mean. Since the median is quite robust, it does not benefit much from outlier removal (at least if the outlier percentage is not extreme). Thus I could imagine, that the entire computation of the correction surface could be done by simply computing the median of the original (corrupted) data.

Response: Thank you for your valuable comment. We agree that the outlier removal in this workflow, which considers the mean ± 1 standard deviation, is not optimal, as you have pointed out. To compare our statistically (weak) approach with a more robust one, such as using the median, as suggested, we generated a new correction surface based on the median and compared it to ours. This comparison was conducted for the six study sites (map sheets) in epoch 1, and the differences are illustrated in the figure and table below.
A visual comparison reveals that most differences between our correction surface and the one derived using only the median are <= one meter, especially within the map sheet, which is the area used for the correction (i.e., at map sheet level). There are some exceptions in areas with a very small number of sample points, which result in larger deviations between the two spline interpolation surfaces. Additionally, we calculated the differences at sample points of the land cover classes "sealed" and "grass/herb" between the DSM and the reference DTM. DSMs were derived with four different approaches: 1.) applying the published correction surface (mean ± 1 standard deviation), 2.) applying a median correction surface with a window size of 50x50 m (median), 3.) a variant of the published version with a window size of 51x51 m and 4.) an uncorrected version. The median and NMAD values of the differences show nearly identical results between the three different correction approaches. This supports your earlier statement that we could have used only the median to derive the correction surface. Although applying the suggested filter at this stage is not feasible, as it would require rerunning the entire DEM processing, this analysis confirms that we do not expect significant differences in the quality of the final product by changing the filter method.

[Figure]

Study site 1 | Study site 2 | Study site 3
Study site 4 | Study site 4a | Study site 6

Value
- -3.966 - -2
- -1.999 - -1
- -0.999 - -0.5
- -0.499 - 0.5
- 0.501 - 1
- 1.001 - 2

| | Differences of DSM - DTM (swissALTI3D) at grass and sealed surface points | Mean | Median | Std | NMAD |
|---|---|---|---|---|---|
| Study site 1 | mean ± 1 Std & moving window 50x50 (as published) | 0.46 | 0.05 | 3.12 | 0.74 |
| | median & moving window 50x50 | 0.37 | -0.02 | 2.87 | 0.76 |
| | mean ± 1 Std & moving window 51x51 | 0.4 | 0 | 2.87 | 0.75 |
| | uncorrected | 0.45 | 0.14 | 3.33 | 1.47 |
| | | | | | |
| Study site 2 | mean ± 1 Std & moving window 50x50 (as published) | 0.68 | 0.08 | 3.81 | 0.77 |
| | median & moving window 50x50 | 0.88 | 0.09 | 4.17 | 0.75 |
| | mean ± 1 Std & moving window 51x51 | 0.7 | 0.1 | 3.71 | 0.74 |
| | uncorrected | 2.81 | 2.69 | 5.45 | 2.73 |
| | | | | | |
| Study site 3 | mean ± 1 Std & moving window 50x50 (as published) | 0.15 | 0.01 | 11.4 | 0.98 |
| | median & moving window 50x50 | 0.16 | 0 | 10.8 | 0.84 |
| | mean ± 1 Std & moving window 51x51 | 0.17 | 0.01 | 10.8 | 0.83 |
| | uncorrected | 0.65 | 0.57 | 12.2 | 1.33 |
| | | | | | |
| Study site 4 | mean ± 1 Std & moving window 50x50 (as published) | -0.28 | -0.01 | 8.57 | 0.92 |
| | median & moving window 50x50 | -0.25 | -0.02 | 6.81 | 0.83 |
| | mean ± 1 Std & moving window 51x51 | -0.2 | 0.02 | 6.81 | 0.82 |
| | uncorrected | -1.2 | -0.16 | 8.89 | 3.01 |
| | | | | | |
| Study site 4a | mean ± 1 Std & moving window 50x50 (as published) | 0.17 | 0.15 | 11.8 | 1.16 |
| | median & moving window 50x50 | 0.22 | 0.19 | 11.3 | 1.08 |
| | mean ± 1 Std & moving window 51x51 | 0.28 | 0.21 | 11.3 | 1.07 |
| | uncorrected | -1.36 | -1.48 | 12.4 | 4.42 |
| | | | | | |
| Study site 5 | mean ± 1 Std & moving window 50x50 (as published) | -5 | 0.11 | 60.5 | 0.97 |
| | median & moving window 50x50 | -3.62 | 0.24 | 53.2 | 1.01 |
| | mean ± 1 Std & moving window 51x51 | -3.85 | 0.24 | 53.2 | 0.94 |
| | uncorrected | -2.95 | 2.22 | 59.7 | 1.54 |

Relation to GSD

The nice thing about photogrammetry is that the accuracy of the result can be linked to the GSD. Of course, the accuracy will depend on the object type: rough surfaces like forests will be worse than smooth objects. Currently, this link is not emphasized at all. One would expect that the accuracy of certain well defined objects (e.g. smooth surfaces) should be around one GSD.

Maybe the age of the images can play some role here, but on the other hand the images are not from WWII and were captured with high-end photogrammetric cameras back in the day. Also the accuracy of the reference data will have some effect.

You are deriving DSMs with grid width three times the GSD. So the accuracy expectation above may not be obtainable everywhere when comparing height models. However, smooth horizontal terrain regions should not be affected by the grid width. Thus reporting the obtained accuracies there (as multiples of the GSD) in the conclusion chapter would show the optimum accuracy. This value also would ease the comparison with other publications that use different data/methods.

Actually, most of this is already done (but just not emphasized); e.g. Fig. 8(a) clearly shows that the NMAD of the horizontal regions is way below 1 m. A quick and dirty estimate from the figure makes one think the NMAD is around 0.7 m, which would correspond to 2*GSD, but I am sure you can report proper numbers.

Response: Thank you for your observation. Following your suggestions, we calculated the NMAD for each epoch, focusing on elevation values below 2000 meters and slope values under 20 ° to exclude mountainous and steep areas. The elevation difference values used are the same as those in Figure 8. As you estimated, the derived accuracy ranges from 0.55 m for epoch 3 to 0.72 m for epoch 4, approximately equal to 2 times the GSD (1.57 and 2.06, respectively), with all epochs assuming a GSD of 0.35 m. The results related to GSD have been added to the conclusion section.

R 529-530. "The vertical accuracy (NMAD) in areas below 2000 m a.s.l. and with a slope of less than 20° ranges from 1.57 to 2.06 times the GSD (on average 0.35 m)."

**Minor detailed comments:**

row 87: "Representative study sites of approximately 210 km2"
 --> in total or is this the area of each site?

Response: Thank you, this was not clear. The area refers to each site and corresponds to one map sheet of the 1:25,000 map sheet units. We clarified this in the text.
R 108-109: "Six representative study sites of approximately 210 km2 each (i.e., one map sheet) were selected within these regions (Fig. 2)".

r89: "consists of approximately 40,000 panchromatic images"
--> it would be interesting to report the number of images per epoch. The histograms in Fig. 2 only allow for a rough estimate. Also, it would be helpful to report the format of the swisstopo tiles. I was not able to find the format using Google (in acceptable time). A rough estimation makes me think it is 17.5 x 12 km².

Response: Thank you for your suggestions. We agree that it is important to report the total number of images per epoch, and we have added this information to the updated version of Figure 2, which has now become Figure 1. Additionally, we noticed an incorrect value in the text regarding the approximate number of images, which we have corrected to 32,000.
Your rough estimation of the dimensions of the swisstopo tiles is correct; they are indeed 12 x 17.5 km. We have included this information in the text as well.
R 88-90: "Each campaign spans approximately 7 years to cover the entire country. Image acquisition was guided by the 1:25000 map sheet units (12 x 17.5 km) representing the topographic maps of Switzerland (swisstopo, 2024b)."

Some information on the orientation quality would be helpful. So far the only figure in this respect is "resulted in accurate horizontal positioning but biases in the Z direction, ranging from 1 to 5 m" in row 123. How accurate is the horizontal positioning?
Perhaps swisstopo provides information like the reference standard deviation of the respective aerial triangulations (comparable to mean reprojection error)? At least, the dedicated publication on the orientation of epoch 2 should provide some values, that can be cited.

Response: This is a good observation, but we have limited information. We added values about planimetric accuracies from the study of Heisig and Simmen (2021). For the orientation of the images from epochs 1, 2 and 4, there is no proper documentation of the orientation process.
R 151-157: "Their study reported planimetric residuals of the GCPs for the different triangulation blocks, which ranged from 2 to 3 pixels (0.6-0.9 m RMSE). Additionally, a comparison between the derived historical orthomosaic and a reference orthomosaic at sample

points evenly distributed over Switzerland revealed a median difference of 1 m, with no systematic bias across Switzerland (Heisig and Simmen, 2021). Although there are no documented values for the planimetric accuracies for the images from epochs 1, 3 and 4, comparisons of the ortho mosaics from those epochs to one derived from epoch 2 show similar results."

r140: "two strategies were adopted, namely urban and low contrast."
--> the reader might wonder why an "urban" strategy is used, although the main focus are forested areas. I think it has to do with the anticipated height changes, which are similar in urban and vegetation area, but a sentence in this respect would be good.
Actually, an information comes later (r148), but it would be good if this would right after introducing the strategies.

Response: Thank you for your observation. Exactly, the "urban" strategy was primarily derive optimal matching results in areas with abrupt height changes, which are a prominent characteristic of forests. We have improved the description of the motivation for using the urban strategy and referenced it earlier in the text.
R 171-174: "The urban strategy utilises internal correlation parameters that account for strong elevation discontinuities, which represents the most prominent characteristics of forest canopies. In contrast, the low-contrast strategy aims to identify congruent points on highly homogeneous surfaces, such as meadows or bare land."

r154: "this correction was also applied to the DSM from epoch 2, resulting in neglectable changes due to the already precise horizontal and vertical orientation."
--> It is interesting that epoch 2 fits much better from the very start. Probably, the ground control information used for the AT of epoch 2 (Heisig & Simmen, 2021) is also based on the very same reference DTM like the here presented correction strategy?

Response: Yes for AT of epoch 2 the DTM SwissALTI3D was used. With the use of abundante GCPs (with derived height values from SwissALTI3D), the absolute orientation leads to stable results over the whole country. For this reason applying the correction workflow to these DSMs had only very little effect. In the orientation process for epoch 1, 3 and 4, which was conducted before swissAlit3D existed, GCPs coordinates were collected with stereo-measurements on already oriented imagery, most probably by different operators.

r161: "which were generalized using the median height"

--> Perhaps, "merged" is a better term than "generalized"? Also, did all the overlapping DSMs from the individual stereo pairs have the same raster layout - meaning same grid width (yes, 1 m) but also same offset (?). If not, some accuracy might get lost during the necessary interpolation for the median-based fusion.

Response: Thank you for the suggestion. We changed the word "generalized" to "merged" as proposed. Yes, the DSMs from the individual stereo pairs do have the same resolution (1 m) and offset. Multiple elevation values were only merged for cells with identical x and y coordinates. We clarified it in the text.

R 185-192: "For each map sheet geometry, all DSMs from the same epoch that corresponded to individual stereo footprints were selected based on spatial intersection. They were then merged and cropped to fit the extent of the map sheet, which typically included around 50 stereo footprints in flat areas and up to 150 in high-alpine regions. Since adjacent stereo footprints spatially overlapped, the pixels in these overlapping areas often included multiple elevation values. To address this, the elevation values were merged using the median height. Grid cells without any successfully matched elevation information were interpolated using a triangular irregular network (TIN) surface."

r166-r170: The description of the correction workflow is not entirely clear to me. Usually, moving windows have an odd number of rows and columns, because this allows to center the window in the central pixel, analyze the data within the window and store the outcome in that central pixel. You, however, use a 50 x 50 moving window. How do you place it on the data raster?

Response: Thank you for this valuable observation. The correction surface was created using ArcGIS, which utilises metrics such as mean and standard deviation (the version used for this study did not provide additional metrics). A 50x50 pixel window was used, which aligns precisely with the input raster. To achieve this alignment, ArcGIS interpolates the mean and standard deviation values for the assigned pixel since there is no exact central pixel. To ensure that using a 50x50 window does not significantly impact the quality of our products, we recalculated the correction surfaces for the six study sites from epoch 1 (as shown in the table above) using a moving window of 51x51 pixels and applied the correction to the DSM. The differences between the DSM and the reference DTM, as indicated in the table, show almost no differences in the observed median and NMAD values. Therefore, while a 50x50m window is not optimal, it does not affect the correction surface.

How does one get to the number "1200" for the regular grid cells per map sheet?

Response: The number of regular grid cells was determined through an empirical approach. This choice represents a tradeoff between obtaining a sufficient number of difference values to derive a valid median for each grid cell (comparing the reference DTM with the DSM on sealed surfaces and grass/herb points) and ensuring enough median cells to create a continuous correction surface. This surface must also respect the spatial variability of the orientation error. We clarified this in the text.

R 210-214: "The median of the remaining elevation differences was then calculated for each of the 1,200 regular grid cells per map sheet. The number of regular grid cells was determined through an empirical approach, ensuring that each grid cell contained enough difference values to yield a valid median and that there were sufficient grid cells per map sheet to enable the interpolation of a continuous correction surface using spline interpolation."

Fig.4: "Point classified as sealed surface and grass/herb" --> "sealed surface or grass/herb"

Response: Thank you for spotting this. We corrected the caption to "sealed surface and grass/herb".

r180: The word "existing" in "existing objects" might be confusing. Either drop it or restrict it to the historical epochs, so that it can not be confused with meaning "existing today".

Response: We dropped the word "existing".
R 227-228: "This process resulted in a normalised DSM (nDSM), representing the heights of all objects above ground."

r185: "nDSM raster cells with values > 60 m were considered outliers"
--> buildings in larger cities may not be larger than 60 m?

Response: Good observation; some buildings may exceed heights of 60 meters. However, our primary focus in this work was to create a vegetation height model. By eliminating pixel values greater than 60 meters, we could reduce the number of unrealistic vegetation heights resulting from erroneous image matching. Since we provide the DSM, the normalised DSM, including buildings taller than 60 meters, can be easily derived. We added a corresponding sentence in the discussion.
R 440-441: "For urban analysis, we recommend using the DSM and normalising it again if needed, as pixel heights larger than 60 m in the VHM were removed."

r187: "To achieve this, a binary non-vegetation mask (0 = non-vegetation and 1 = vegetation) was created"
--> Wouldn't setting non-vegetation to NaN be a better choice in order to keep proper vegetation differences, which are exactly 0? i.e. after the multiplication with this mask (row 192) a pixel with value == 0 can not be told apart from a proper vegetation pixel with difference == 0 and a masked non-vegetation pixel.

Response: Thank you for the observation. Using NaN for non-vegetated areas may be beneficial when calculating the difference between two vegetation height models. If the difference in vegetated pixels is exactly zero, the current masking method does not allow us to distinguish these pixels from non-vegetated ones. However, setting non-vegetated pixels to NaN could lead to complications. Specifically, if vegetation exists in only one of the height models at a specific location, this could hinder the differencing process and affect the accuracy of the extracted vegetation heights for that area.

r190: "To retain vegetation close to buildings and streets in the VHMs, single trees and hedges from the TLM were buffered by 15 m"
--> Following this 15m-buffering, some parts of buildings may end up as being considered as vegetation?

Response: Yes, this is true. Our historical vegetation height models cannot be easily applied to urban areas in an automated way. However, this approach still captures some significant vegetation in urban settings, which could be valuable for more detailed analysis.

r220: "For each forest plot, 25 elevation measurements were extracted on the **same** stereo pair as used for generating the historical DSM"
--> But the historic DSM in the end is created from a median based fusion, where the DSM from this "single" stereo pair is no longer "available". So, manual image stereo measurements from one image pair are not really compared with automatic heights from the very same image pair?

Response: Thank you for your comment. It is true that the final elevation is the median of overlapping stereo pairs. We replaced "the same stereo pair" with "one of the overlapping stereo pairs" and added one sentence to the discussion.
R 481-484: "Also, the measurement was only performed on one stereo model, while the DSM in this study was created from the fusion of several overlapping pairs."

r224: Regarding "completeness": A differentiation regarding how a pixel got its height is not performed, meaning from building strategy or from low contrast strategy or from median-based fusion?

Response: This would be interesting; however, we did not conduct this analysis, and we measured the percentage of pixels with a successful match using either strategy.

Fig 6: I am confused by the total number of grid points reported for the six classes. They add up in total to 226606 points (=6101+43654+77963+13488+74065+11335). However (if I am not mistaken), the six considered tiles of size 210 km² would result only in 6 * 210 / 0.1^2 = 126000 points, where the spacing of the points is 100 m (according to row 201). How can this huge difference be explained?

Response: Your observation is correct. The higher number comes from conducting the completeness analysis on the entire DSM extent before cropping it to the map sheet boundaries. This was done by merging all individual stereo DSMs that intersect with the map sheet geometry. While this approach does not align perfectly with the size of the study sites, we believe it enhances the robustness of the calculated mean completeness by including more data points. To clarify this situation, we add a sentence in Section 2.3.2.
R 279-280: "Completeness values were calculated for all four epochs from the DSMs of the six study sites before cropping to the actual map sheet extent (see section 2.2.2)."

r281 "and an NMAD of approximately 1.5 m."
--> The respective number in table 2 is actually 1.41 m. So rounding would be 1.4 m, although I'd prefer to cite the exact number (i.e. 1.41). (This may seem a bit nitpicking, but given the many numbers in different figures and tables it would be helpful to keep the very same digits in order to help the reader pricely located quoted values.)

Response: Thank you for this recommendation. We reported the correct values in the text without rounding.

Table 2: Interestingly, the smallest error values result for epoch 1. Which appears a bit surprising, considering that this is the oldest epoch, thus being effect by long storage harm, etc. the most (maybe even being captured by the least advanced camera?). Any explanations for this? Maybe epoch 1 has better GSD or better forward/side overlap?

Response: Thank you for your observation. As one possible explanation for this observation we consider the strong expansion of settled areas and changes in forest cover over the whole time period. Knowing that image matching results are affected close to buildings and forests, we recalculated the differences between the DSM and DTM on sealed surface points, excluding all points closer than 10 m to buildings or forests. The observed decrease in accuracy over time is now not present anymore. The statistics (see updated Table 2) indicate a slightly better accuracy for epoch 3, although the differences between the median and NMAD values among epochs are at the centimetre level.

But maybe the picture changes after adopting the robust statistics mentioned in the beginning?

Response: We may not have understood the comment correctly. The robust statistics for NMAD and the median were calculated before the filtering. The filtering was implemented to derive the RMSE, which is a common statistical metric, but is more sensitive to outliers. As a result, the findings remain the same, apart from the update regarding removing suboptimal sealed surface points.

Fig. 8: Please, add (°) to the two aspect captions.

Response: Thank you for noticing. We have added it to the figure.

r341 "between epoch 4 and 2022 (Fig. 10)."
--> Do you mean a certain part in area 4? then perhaps add the location along the "length", i.e. from 370 m to 870 m.

Response: Good observation. In the text, we added the location in meters, where a cut or damage occurred, and regrowth between epochs 3 and 4 along profile 2 can be detected in the most recent (2022) VHM.
R 408-409: "However, forest growth, cuts or damage followed by regrowth can be detected, for example, between epoch 3 and 4 (cut or damage) and 2022 (regrowth) (Fig. 10b, profile 2, 1250 m - 1500 m)."

Fig. 11. The heading of the center figure says "1994", thus coming from epoch 3 not epoch 2?

Response: Thank you for spotting this mistake. We changed it in the caption of Figure 11. In the text (section 3.4), the correct epoch was reported.

R 431: Figure 11: "Small extent of region 2 (Swiss Plateau), showing the vegetation height models (VHMs) of epochs 1, 3 and 4, overlaid by…"

r382: The completeness of the image-matching shows a steady increase over the four epochs (Fig. A2), particularly in areas classified as grasses and herbs.
 --> Is this increase really so dramatic, and worth mentioning? Some values actually decrease. Or do you mean with respect to epoch 1 (which, however, is not included in the referenced Fig. A2).

Response: Thank you for your observation. The increase in completeness is associated with epoch 1 (which is correctly only shown in Figure 6). However, this increase primarily applies to the classes "Grass" and "Glacial and Perpetual Snow" and not all classes. We have improved the description of the results in the manuscript.
R 452-454: "The completeness of the image-matching across epochs remains nearly constant (Fig. A12). However, an improvement in completeness is observed for grass and herbs, as well as glacial and perennial snow classes, in relation to Epoch 1, although this may vary depending on the study area."

r383: "This improvement is strongly related to advancements in camera technology ..."
 --> Since the error values for epoch 1 are the smallest in Tab. 2 (currently), this claim stands on weak feet or at least needs some clarification.

Response: Thank you for your observation. As mentioned earlier, the improvement in completeness is related to epoch 1, and only to certain classes. However, since we cannot confirm that advancements in camera technology are the relevant factor contributing to this improvement, we have removed this statement from the revised text.

r386: "at sealed surface points shows metric to sub-metric accuracy with an uncertainty"
--> Somewhere here it should be repeated that the GSD is around 35 cm and that the DSMs were generated with 1 m grid width. I am not a native speaker, but I am not sure, if the term "sub-metric" exists. Perhaps, "meter to sub-meter" is more correct.

Response: Thank you for this observation. We corrected the term and included the GSD and DSMs resolution.
R 459: "... at sealed surface points shows sub-meter values with a maximum NMAD of 0.86 m. This value is roughly 2.5 times the GSD, although the DSMs were generated at 1 m resolution."

r405: "The larger bias in historical data might be due to the more demanding visual detection of objects during manual stereo measurements on panchromatic imagery and the coarser resolution of the images."

--> Actually, the considered historical images in your study have a GSD of approx. 0.35 m, which (nominally) is better then the 0.5 m of the mentioned ADS images.

Response: Thank you for this observation. Detecting objects, such as canopy tops, in stereoscopic measurements can be significantly more challenging with older panchromatic 8-bit images compared to newer near-infrared 12-bit imagery. However, as you noted, the GSD of ADS images is coarser than that of historical images. We have revised the sentence and removed the reference to resolution dependency, as the overall image quality might have a greater impact than image resolution.

R 481-484: Compared to this study, the greater bias in the historical data could be due to the more challenging visual capture of canopy objects in manual stereo measurements on 8-bit panchromatic images. Also, the measurement was only performed on one stereo model, while the DSM in this study was created from the fusion of several overlapping pairs..

r412: "the low quality of scanning film negatives"

--> Just the wording is maybe odd: The scanning itself is not the problem, but the quality of the original films.

Response: Thank you. This is not the case for the Swiss historical images, as they are scanned with a photogrammetric scanner, and considerable attention has been paid to the quality of the scanning. However, datasets from other archives may suffer from low scanning quality, such as the introduction of scanning stripes, among other issues. We have clarified the sentence.

R 490-491: "… the high cost of images, deteriorated film negatives and the poor …"

r422: "vertical correction of about 20,000 DSMs"

--> Maybe add "(stereo footprints)".

Response: Thank you for the suggestion. Our wording is misleading, and we corrected it.

R 501-502: "To handle countrywide datasets acquired over three decades, the processing of approximately 32,000 images, as well as the post-processing, including the vertical correction of about 20,000 stereo footprint DSMs, was done at the map sheet level."

---

## Author Response (AR2)

**Title: Countrywide Digital Surface Models and Vegetation Height Models from Historical Aerial Images**

Author(s): Mauro Marty et al.

Handling editor: Alexander Gruber, alexander.gruber@geo.tuwien.ac.at

**Author's response related to the comments made in Report #1 received on the manuscript: <a href="https://doi.org/10.5194/essd-2024-428">https://doi.org/10.5194/essd-2024-428</a>**

We thank anonymous referee #2 for some remaining recommendations concerning clarity and presentation of our manuscript. After thoroughly evaluating the feedback and recommendations, we have revised our manuscript to further enhance the clarity and accuracy of our paper.

The main changes we made to the manuscript are as follows:

- Incorporating the suggestions into the figures.
- Modifying certain formulations-within the text.

We are confident that the following point-by-point responses adequately address the remaining suggestions.

**Point-by-point reply to reviewer comments**: Reviewers' comments are in black, and the authors' response is in blue, with citations from the revised manuscript in green.

**RC 2 – Report # 1:**

- r128 "the images from epochs 1, 2 and 4 was derived by swisstopo"
- --> "1, 3 and 4"

Response: Thank you very much for spotting this mistake. We have changed the wrong campaign numbers from "1, 2 and 4" to "1, 3 and 4" in the sentence.

R127-129: The exterior orientation information for the images from epochs 1, 3 and 4 was derived by swisstopo within the framework of the Swiss Land Use/Cover Statistics program around 2004.

- Fig. 3: I was a bit puzzled by the bold black corner markers. It took me some time to find out that they may enclose the two workflows for DSM and VHM. Maybe you could improve that? e.g. by surrounding them with slightly colored rectangles? And/or add to the caption that the DSM workflow involves the steps 2-5 and the VHM afterwards additionally 6-8.

Response: Thank you, we have improved the figure 3 as suggested.

- r183: Thanks for the added info regarding the "1200 regular grid cells". However, I still think there is room for improvement, because I do not think that the "number of cells" is the typical information regarding a grid layout, but the grid width. Thus I would suggest to replace "The number of regular grid cells was" by "The cell size of this regular grid (XXX m) was", where XXX is the grid width. (Is it around 418 m = sqrt(12\*17.5/1200)?) And add this cell size also to the description in Fig 4, because currently the 1st legend entry "Grid" is difficult to link to the description in the main text.

Response: Thank you for this comment. We replaced the number of grid cells by the grid cell size and rephrased the caption for figure 4 correspondingly.

R 185-188: The cell size of this regular grid (418 m) was determined through an empirical approach, ensuring that each grid cell contained enough difference values to yield a valid median and that there were sufficient grid cells per map sheet to enable the interpolation of a continuous correction surface using spline interpolation.

R:1991-195: Figure 4: Illustration of the workflow to correct the elevation bias in the historical digital surface models (DSMs) for each map sheet. A correction surface is derived by a spline interpolation of the median height difference values (reference DTM – historical DSM at sample points "sealed surface" and "grass/herbs") within regular grid cells (418 m).

Additionally, you could adapt the wording such that the outlier removal was not done only by the filtering part using mean +- 1 std, but also by considering the median in the 2nd step. Currently, it reads like the outlier removal was done only by the mean+std part (which as outlined in my 1st review is an odd way); e.g. "Outliers were dealt with in the following way: First, the height differences were filtered based on ... 50 x 50 pixels. Afterwards, the median of the remaining ..."

Response: We appreciate your recommendation for enhancement. We have incorporated your suggestion into the text, agreeing that this formulation supports that the outlier removal considers robust statistics, by calculating the median of the oddly filtered height differences.

R 182-185: The approach to handle outliers was as follows: In a first step, the height differences were filtered based on mean  $\pm$  1 standard deviation (Std), calculated within a moving window of  $50 \times 50$  pixels. In a second step the median of the remaining elevation differences was calculated for each of the 1,200 regular grid cells per map sheet.

- r262: "such as the median and the normalised median absolute deviation (NMAD) were biases spatial often distributed" calculated. as in datasets are not normally --> I find that 2nd part confusing, because the valid application of the NMAD >>requires<< that the underlying distribution is in fact Gaussian. That's the reason why the factor from MAD to NMAD is 1.4826. For a different distribution a different factor would be needed. Perhaps, that 2nd part of the sentence is not referring to the applicability of the NMAD and really just means the "bias". But the bias itself has no distribution as it is just an (unknown) offset. Perhaps, you can reformulated (or drop that 2nd part).

Response: Thank you for this comment. We agree that a bias does not have a distribution. The formulation of the mentioned sentence is indeed confusing. We document the variation of the distribution with the NMAD, because the NMAD is an estimate of the standard deviation more resilient to outliers. We have omitted the 2nd part of our sentence, as we believe that it remains clear what we accomplished, and a detailed explanation would not be appropriate at this position in the text.

R 265-266: From the remaining differences, robust statistics, such as the median and the normalised median absolute deviation (NMAD) were calculated (Höhle and Höhle, 2009).

- Fig. 7: Note, that the limits for the histograms, probably, could/should be set to +-5 m, instead of +-10 m, in order to reveal some details.

Response: Thank you for this suggestion. We have changed the x limit to +-5 m and updated the figure accordingly.

- Fig. 9: On the left we see that the profile runs diagonal through exactly 5 blue points. Also, the right side shows 5 blue points, however, the distance between them is not identical anymore.

Response: Thank you for spotting this mistake. We have corrected the plot and updated the figure accordingly.

- Fig. 10: Probably, the five colors of the VHMs could be changed to something more discernible?

Response: Thank you for your suggestion. We tested other colour maps; however, the visual appearance does not improve, and we prefer to maintain the current version. In this version, classes 0-1 are clearly visible, followed by a gradual transition to light and dark green representing vegetation height. In addition, by keeping the current colour scale, we are consistent with the VHM map displayed in Figure 11.